# Asymptotic Guarantees for Learning Generative Models with the Sliced-Wasserstein Distance

**Kimia Nadjahi**[1], **Alain Durmus**[2], **Umut Şimşekli**[1,3], **Roland Badeau**[1]
1: LTCI, Télécom Paris, Institut Polytechnique de Paris, France
2: CMLA, ENS Cachan, CNRS, Université Paris-Saclay, France
3: Department of Statistics, University of Oxford, UK
{kimia.nadjahi, umut.simsekli, roland.badeau}@telecom-paris.fr
alain.durmus@cmla.ens-cachan.fr

## Abstract

Minimum expected distance estimation (MEDE) algorithms have been widely used for probabilistic models with intractable likelihood functions and they have become increasingly popular due to their use in implicit generative modeling (e.g. Wasserstein generative adversarial networks, Wasserstein autoencoders). Emerging from computational optimal transport, the Sliced-Wasserstein (SW) distance has become a popular choice in MEDE thanks to its simplicity and computational benefits. While several studies have reported empirical success on generative modeling with SW, the theoretical properties of such estimators have not yet been established. In this study, we investigate the asymptotic properties of estimators that are obtained by minimizing SW. We first show that convergence in SW implies weak convergence of probability measures in general Wasserstein spaces. Then we show that estimators obtained by minimizing SW (and also an approximate version of SW) are asymptotically consistent. We finally prove a central limit theorem, which characterizes the asymptotic distribution of the estimators and establish a convergence rate of $\sqrt{n}$, where $n$ denotes the number of observed data points. We illustrate the validity of our theory on both synthetic data and neural networks.

## 1 Introduction

Minimum distance estimation (MDE) is a generalization of maximum-likelihood inference, where the goal is to minimize a distance between the empirical distribution of a set of independent and identically distributed (i.i.d.) observations $Y_{1:n} = (Y_1, \ldots, Y_n)$ and a family of distributions indexed by a parameter $\theta$. The problem is formally defined as follows [1, 2]:

$$\hat{\theta}_n = \mathrm{argmin}_{\theta \in \Theta} \mathbf{D}(\hat{\mu}_n, \mu_\theta) , \tag{1}$$

where $\mathbf{D}$ denotes a distance (or a divergence in general) between probability measures, $\mu_\theta$ denotes a probability measure indexed by $\theta$, $\Theta$ denotes the parameter space, and

$$\hat{\mu}_n = \frac{1}{n} \sum_{i=1}^{n} \delta_{Y_i} \tag{2}$$

denotes the empirical measure of $Y_{1:n}$, with $\delta_Y$ being the Dirac distribution with mass on the point $Y$. When $\mathbf{D}$ is chosen as the Kullback-Leibler divergence, this formulation coincides with the maximum likelihood estimation (MLE) [2].

While MDE provides a fruitful framework for statistical inference, when working with generative models, solving the optimization problem in (1) might be intractable since it might be impossible to evaluate the probability density function associated with $\mu_\theta$. Nevertheless, in various settings, even if

the density is not available, one can still generate samples from the distribution $\mu_\theta$, and such samples turn out to be useful for making inference. More precisely, under such settings, a natural alternative to (1) is the minimum *expected* distance estimator, which is defined as follows [3]:

$$\hat{\theta}_{n,m} = \operatorname{argmin}_{\theta \in \Theta} \mathbb{E}\left[\mathbf{D}(\hat{\mu}_n, \hat{\mu}_{\theta,m})|Y_{1:n}\right] . \tag{3}$$

Here,

$$\hat{\mu}_{\theta,m} = \frac{1}{m} \sum\nolimits_{i=1}^{m} \delta_{Z_i} \tag{4}$$

denotes the empirical distribution of $Z_{1:m}$, that is a sequence of i.i.d. random variables with distribution $\mu_\theta$. This algorithmic framework has computationally favorable properties since one can replace the expectation with a simple Monte Carlo average in practical applications.

In the context of MDE, distances that are based on optimal transport (OT) have become increasingly popular due to their computational and theoretical properties [4, 5, 6, 7, 8]. For instance, if we replace the distance $\mathbf{D}$ in (3) with the Wasserstein distance (defined in Section 2 below), we obtain the minimum expected Wasserstein estimator [3]. In the classical statistical inference setting, the typical use of such an estimator is to infer the parameters of a measure whose density does not admit an analytical closed-form formula [2]. On the other hand, in the implicit generative modeling (IGM) setting, this estimator forms the basis of two popular IGM strategies: Wasserstein generative adversarial networks (GAN) [4] and Wasserstein variational auto-encoders (VAE) [5] (cf. [9] for their relation). The goal of these two methods is to find the best parametric *transport map* $T_\theta$, such that $T_\theta$ transforms a simple distribution $\mu$ (e.g. standard Gaussian or uniform) to a potentially complicated data distribution $\hat{\mu}_n$ by minimizing the Wasserstein distance between the transported distribution $\mu_\theta = T_{\theta\sharp}\mu$ and $\hat{\mu}_n$, where $\sharp$ denotes the push-forward operator, to be defined in the next section. In practice, $\theta$ is typically chosen as a neural network, for which it is often impossible to evaluate the induced density $\mu_\theta$. However, one can easily generate samples from $\mu_\theta$ by first generating a sample from $\mu$ and then applying $T_\theta$ to that sample, making minimum expected distance estimation (3) feasible for this setting. Motivated by its practical success, the theoretical properties of this estimator have been recently taken under investigation [10, 11] and very recently Bernton et al. [3] have established the consistency (for the general setting) and the asymptotic distribution (for one dimensional setting) of this estimator.

Even though estimation with the Wasserstein distance has served as a fertile ground for many generative modeling applications, except for the case when the measures are supported on $\mathbb{R}^1$, the computational complexity of minimum Wasserstein estimators rapidly becomes excessive with the increasing problem dimension, and developing accurate and efficient approximations is a highly non-trivial task. Therefore, there have been several attempts to use more practical alternatives to the Wasserstein distance [12, 6]. In this context, the Sliced-Wasserstein (SW) distance [13, 14, 15] has been an increasingly popular alternative to the Wasserstein distance, which is defined as an average of *one-dimensional* Wasserstein distances, which allows it to be computed in an efficient manner.

While several studies have reported empirical success on generative modeling with SW [16, 17, 18, 19], the theoretical properties of such estimators have not yet been fully established. Bonnotte [14] proved that SW is a proper metric, and in *compact* domains SW is equivalent to the Wasserstein distance, hence convergence in SW implies weak convergence in compact domains. [14] also analyzed the gradient flows based on SW, which then served as a basis for a recently proposed IGM algorithm [18]. Finally, recent studies [16, 20] investigated the sample complexity of SW and established bounds for the SW distance between two measures and their empirical instantiations.

In this paper, we investigate the asymptotic properties of estimators given in (1) and (3) when $\mathbf{D}$ is replaced with the SW distance. We first prove that convergence in SW implies weak convergence of probability measures defined on general domains, which generalizes the results given in [14]. Then, by using similar techniques to the ones given in [3], we show that the estimators defined by (1) and (3) are consistent, meaning that as the number of observations $n$ increases the estimates will get closer to the data-generating parameters. We finally prove a central limit theorem (CLT) in the multidimensional setting, which characterizes the asymptotic distribution of these estimators and establishes a convergence rate of $\sqrt{n}$. The CLT that we prove is stronger than the one given in [3] in the sense that it is not restricted to the one-dimensional setting as opposed to [3].

We support our theory with experiments that are conducted on both synthetic and real data. We first consider a more classical statistical inference setting, where we consider a Gaussian model and a

multidimensional $\alpha$-stable model whose density is not available in closed-form. In both models, the experiments validate our consistency and CLT results. We further observe that, especially for high-dimensional problems, the estimators obtained by minimizing SW have significantly better computational properties when compared to the ones obtained by minimizing the Wasserstein distance, as expected. In the IGM setting, we consider the neural network-based generative modeling algorithm proposed in [16] and show that our results also hold in the real data setting as well.

## 2 Preliminaries and Technical Background

We consider a probability space $(\Omega, \mathcal{F}, \mathbb{P})$ with associated expectation operator $\mathbb{E}$, on which all the random variables are defined. Let $(Y_k)_{k \in \mathbb{N}}$ be a sequence of random variables associated with observations, where each observation takes value in $\mathsf{Y} \subset \mathbb{R}^d$. We assume that these observations are i.i.d. according to $\mu_\star \in \mathcal{P}(\mathsf{Y})$, where $\mathcal{P}(\mathsf{Y})$ stands for the set of probability measures on $\mathsf{Y}$.

A statistical model is a family of distributions on $\mathsf{Y}$ and is denoted by $\mathcal{M} = \{\mu_\theta \in \mathcal{P}(\mathsf{Y}), \ \theta \in \Theta\}$, where $\Theta \subset \mathbb{R}^{d_\theta}$ is the parametric space. In this paper, we focus on parameter inference for purely generative models: for all $\theta \in \Theta$, we can generate i.i.d. samples $(Z_k)_{k \in \mathbb{N}^*} \in \mathsf{Y}^{\mathbb{N}^*}$ from $\mu_\theta$, but the associated likelihood is numerically intractable. In the sequel, $(Z_k)_{k \in \mathbb{N}^*}$ denotes an i.i.d. sequence from $\mu_\theta$ with $\theta \in \Theta$, and for any $m \in \mathbb{N}^*$, $\hat{\mu}_{\theta,m} = (1/m) \sum_{i=1}^{m} \delta_{Z_i}$ denotes the corresponding empirical distribution.

Throughout our study, we assume that the following conditions hold: (1) $\mathsf{Y}$, endowed with the Euclidean distance $\rho$, is a Polish space, (2) $\Theta$, endowed with the distance $\rho_\Theta$, is a Polish space, (3) $\Theta$ is a $\sigma$-compact space, *i.e.* the union of countably many compact subspaces, and (4) parameters are identifiable, *i.e.* $\mu_\theta = \mu_{\theta'}$ implies $\theta = \theta'$. We endow $\mathcal{P}(\mathsf{Y})$ with the Lévy-Prokhorov distance $\mathbf{d}_\mathcal{P}$, which metrizes the weak convergence by [21, Theorem 6.8] since $\mathsf{Y}$ is assumed to be a Polish space. We denote by $\mathcal{Y}$ the Borel $\sigma$-field of $(\mathsf{Y}, \rho)$.

**Wasserstein distance.** For $p \geq 1$, we denote by $\mathcal{P}_p(\mathsf{Y})$ the set of probability measures on $\mathsf{Y}$ with finite $p$'th moment: $\mathcal{P}_p(\mathsf{Y}) = \left\{ \mu \in \mathcal{P}(\mathsf{Y}) : \int_\mathsf{Y} \|y - y_0\|^p \, \mathrm{d}\mu(y) < +\infty, \text{ for some } y_0 \in \mathsf{Y} \right\}$. The Wasserstein distance of order $p$ between any $\mu, \nu \in \mathcal{P}_p(\mathsf{Y})$ is defined by [22],

$$\mathbf{W}_p^p(\mu, \nu) = \inf_{\gamma \in \Gamma(\mu,\nu)} \left\{ \int_{\mathsf{Y} \times \mathsf{Y}} \|x - y\|^p \, \mathrm{d}\gamma(x,y) \right\} , \tag{5}$$

where $\Gamma(\mu, \nu)$ is the set of probability measures $\gamma$ on $(\mathsf{Y} \times \mathsf{Y}, \mathcal{Y} \otimes \mathcal{Y})$ satisfying $\gamma(\mathsf{A} \times \mathsf{Y}) = \mu(\mathsf{A})$ and $\gamma(\mathsf{Y} \times \mathsf{A}) = \nu(\mathsf{A})$ for any $\mathsf{A} \in \mathcal{B}(\mathsf{Y})$. The space $\mathcal{P}_p(\mathsf{Y})$ endowed with the distance $\mathbf{W}_p$ is a Polish space by [22, Theorem 6.18] since $(\mathsf{Y}, \rho)$ is assumed to be Polish.

The one-dimensional case is a favorable scenario for which computing the Wasserstein distance of order $p$ between $\mu, \nu \in \mathcal{P}_p(\mathbb{R})$ becomes relatively easy since it has a closed-form formula, given by [23, Theorem 3.1.2.(a)]:

$$\mathbf{W}_p^p(\mu, \nu) = \int_0^1 \left| F_\mu^{-1}(t) - F_\nu^{-1}(t) \right|^p \mathrm{d}t = \int_\mathbb{R} \left| s - F_\nu^{-1}(F_\mu(s)) \right|^p \mathrm{d}\mu(s) , \tag{6}$$

where $F_\mu$ and $F_\nu$ denote the cumulative distribution functions (CDF) of $\mu$ and $\nu$ respectively, and $F_\mu^{-1}$ and $F_\nu^{-1}$ are the quantile functions of $\mu$ and $\nu$ respectively. For empirical distributions, (6) is calculated by simply sorting the $n$ samples drawn from each distribution and computing the average cost between the sorted samples.

**Sliced-Wasserstein distance.** The analytical form of the Wasserstein distance for one-dimensional distributions is an attractive property that gives rise to an alternative metric referred to as the Sliced-Wasserstein (SW) distance [13, 15]. The idea behind SW is to first, obtain a family of one-dimensional representations for a higher-dimensional probability distribution through linear projections, and then, compute the average of the Wasserstein distance between these one-dimensional representations.

More formally, let $\mathbb{S}^{d-1} = \left\{ u \in \mathbb{R}^d : \|u\| = 1 \right\}$ be the $d$-dimensional unit sphere, and denote by $\langle \cdot, \cdot \rangle$ the Euclidean inner-product. For any $u \in \mathbb{S}$, we define $u^\star$ the linear form associated with $u$ for any $y \in \mathsf{Y}$ by $u^\star(y) = \langle u, y \rangle$. The Sliced-Wasserstein distance of order $p$ is defined for any $\mu, \nu \in \mathcal{P}_p(\mathsf{Y})$ as,

$$\mathbf{SW}_p^p(\mu, \nu) = \int_{\mathbb{S}^{d-1}} \mathbf{W}_p^p(u_\sharp^\star \mu, u_\sharp^\star \nu) \mathrm{d}\boldsymbol{\sigma}(u) \tag{7}$$

where $\boldsymbol{\sigma}$ is the uniform distribution on $\mathbb{S}^{d-1}$ and for any measurable function $f : \mathsf{Y} \to \mathbb{R}$ and $\zeta \in \mathcal{P}(\mathsf{Y})$, $f_\sharp \zeta$ is the push-forward measure of $\zeta$ by $f$, *i.e.* for any $\mathsf{A} \in \mathcal{B}(\mathbb{R})$, $f_\sharp \zeta(\mathsf{A}) = \zeta(f^{-1}(\mathsf{A}))$ where $f^{-1}(\mathsf{A}) = \{y \in \mathsf{Y} : f(y) \in \mathsf{A}\}$.

$\mathbf{SW}_p$ is a distance on $\mathcal{P}_p(\mathsf{Y})$ [14] and has important practical implications: in practice, the integration in (7) is approximated using a Monte Carlo scheme that randomly draws a finite set of samples from $\boldsymbol{\sigma}$ on $\mathbb{S}^{d-1}$ and replaces the integral with a finite-sample average. Therefore, the evaluation of the SW distance between $\mu, \nu \in \mathcal{P}_p(\mathsf{Y})$ has significantly lower computational requirements than the Wasserstein distance, since it consists in solving several one-dimensional optimal transport problems, which have closed-form solutions.

## 3 Asymptotic Guarantees for Minimum Sliced-Wasserstein Estimators

We define the *minimum Sliced-Wasserstein estimator* (MSWE) *of order $p$* as the estimator obtained by plugging $\mathbf{SW}_p$ in place of $\mathbf{D}$ in (1). Similarly, we define the *minimum expected Sliced-Wasserstein estimator* (MESWE) *of order $p$* as the estimator obtained by plugging $\mathbf{SW}_p$ in place of $\mathbf{D}$ in (3). In the rest of the paper, MSWE and MESWE will be denoted by $\hat{\theta}_n$ and $\hat{\theta}_{n,m}$ respectively.

We present the asymptotic properties that we derived for MSWE and MESWE, namely their existence and consistency. We study their measurability in Section 2.2 of the supplementary document. We also formulate a CLT that characterizes the asymptotic distribution of MSWE and establishes a convergence rate for any dimension. We provide all the proofs in Section 3 of the supplementary document. Note that, since the Sliced-Wasserstein distance is an average of one-dimensional Wasserstein distances, some proofs are, inevitably, similar to the proofs done in [3]. However, the adaptation of these techniques to the SW case is made possible by the identification of novel properties regarding the topology induced by the SW distance, to the best of our knowledge, which we establish for the first time in this study.

### 3.1 Topology induced by the Sliced-Wasserstein distance

We begin this section by a useful result which we believe is interesting on its own and implies that the topology induced by $\mathbf{SW}_p$ on $\mathcal{P}_p(\mathbb{R}^d)$ is finer than the weak topology induced by the Lévy-Prokhorov metric $\mathbf{d}_\mathcal{P}$.

**Theorem 1.** *Let $p \in [1, +\infty)$. The convergence in $\mathbf{SW}_p$ implies the weak convergence in $\mathcal{P}(\mathbb{R}^d)$. In other words, if $(\mu_k)_{k \in \mathbb{N}}$ is a sequence of measures in $\mathcal{P}_p(\mathbb{R}^d)$ satisfying $\lim_{k \to +\infty} \mathbf{SW}_p(\mu_k, \mu) = 0$, with $\mu \in \mathcal{P}_p(\mathbb{R}^d)$, then $(\mu_k)_{k \in \mathbb{N}} \xrightarrow{w} \mu$.*

The property that convergence in $\mathbf{SW}_p$ implies weak convergence has already been proven in [14] only for *compact* domains. While the implication of weak convergence is one of the most crucial requirements that a distance metric should satisfy, to the best of our knowledge, this implication has not been proved for general domains before. In [14], the main proof technique was based on showing that $\mathbf{SW}_p$ is equivalent to $\mathbf{W}_p$ in compact domains, whereas we follow a different path and use the Lévy characterization.

### 3.2 Existence and consistency of MSWE and MESWE

In our next set of results, we will show that both MSWE and MESWE are consistent, in the sense that, when the number of observations $n$ increases, the estimators will converge to a parameter $\theta_\star$ that minimizes the ideal problem $\theta \mapsto \mathbf{SW}_p(\mu_\star, \mu_\theta)$. Before we make this argument more precise, let us first present the assumptions that will imply our results.

**A1.** *The map $\theta \mapsto \mu_\theta$ is continuous from $(\Theta, \rho_\Theta)$ to $(\mathcal{P}(\mathsf{Y}), \mathbf{d}_\mathcal{P})$, i.e. for any sequence $(\theta_n)_{n \in \mathbb{N}}$ in $\Theta$, satisfying $\lim_{n \to +\infty} \rho_\Theta(\theta_n, \theta) = 0$, we have $(\mu_{\theta_n})_{n \in \mathbb{N}} \xrightarrow{w} \mu_\theta$.*

**A2.** *The data-generating process is such that $\lim_{n \to +\infty} \mathbf{SW}_p(\hat{\mu}_n, \mu_\star) = 0$, $\mathbb{P}$-almost surely.*

**A3.** *There exists $\epsilon > 0$, such that setting $\epsilon_\star = \inf_{\theta \in \Theta} \mathbf{SW}_p(\mu_\star, \mu_\theta)$, the set $\Theta^\star_\epsilon = \{\theta \in \Theta : \mathbf{SW}_p(\mu_\star, \mu_\theta) \leq \epsilon_\star + \epsilon\}$ is bounded.*

These assumptions are mostly related to the identifiability of the statistical model and the regularity of the data generating process. They are arguably mild assumptions, analogous to those that have

already been considered in the literature [3]. Note that, without Theorem 1, the formulation and use of **A**2 in our proofs in the supplementary document would not be possible. In the next result, we establish the consistency of MSWE.

**Theorem 2** (Existence and consistency of MSWE). *Assume A1, A2 and A3. There exists* $\mathsf{E} \in \mathcal{F}$ *with* $\mathbb{P}(\mathsf{E}) = 1$ *such that, for all* $\omega \in \mathsf{E}$,

$$\lim_{n \to +\infty} \inf_{\theta \in \Theta} \mathbf{SW}_p(\hat{\mu}_n(\omega), \mu_\theta) = \inf_{\theta \in \Theta} \mathbf{SW}_p(\mu_\star, \mu_\theta), \quad and \tag{8}$$

$$\limsup_{n \to +\infty} \mathrm{argmin}_{\theta \in \Theta} \mathbf{SW}_p(\hat{\mu}_n(\omega), \mu_\theta) \subset \mathrm{argmin}_{\theta \in \Theta} \mathbf{SW}_p(\mu_\star, \mu_\theta), \tag{9}$$

*where* $\hat{\mu}_n$ *is defined by* (2). *Besides, for all* $\omega \in \mathsf{E}$, *there exists* $n(\omega)$ *such that, for all* $n \geq n(\omega)$, *the set* $\mathrm{argmin}_{\theta \in \Theta} \mathbf{SW}_p(\hat{\mu}_n(\omega), \mu_\theta)$ *is non-empty.*

Our proof technique is similar to the one given in [3]. This result shows that, when the number of observations goes to infinity, the estimate $\hat{\theta}_n$ will converge to a global minimizer of the problem $\min_{\theta \in \Theta} \mathbf{SW}_p(\mu_\star, \mu_\theta)$.

In our next result, we prove a similar property for MESWEs as $\min(m, n)$ goes to infinity. In order to increase clarity, and without loss of generality, in this setting, we consider $m$ as a function of $n$ such that $\lim_{n \to +\infty} m(n) = +\infty$. Now, we derive an analogous version of Theorem 2 for MESWE. For this result, we need to introduce another continuity assumption.

**A4.** *If* $\lim_{n \to +\infty} \rho_\Theta(\theta_n, \theta) = 0$, *then* $\lim_{n \to +\infty} \mathbb{E}[\mathbf{SW}_p(\mu_{\theta_n}, \hat{\mu}_{\theta_n, n}) | Y_{1:n}] = 0$.

The next theorem establishes the consistency of MESWE.

**Theorem 3** (Existence and consistency of MESWE). *Assume A1, A2, A3 and A4. Let* $(m(n))_{n \in \mathbb{N}^*}$ *be an increasing sequence satisfying* $\lim_{n \to +\infty} m(n) = +\infty$. *There exists a set* $\mathsf{E} \subset \Omega$ *with* $\mathbb{P}(\mathsf{E}) = 1$ *such that, for all* $w \in \mathsf{E}$,

$$\lim_{n \to +\infty} \inf_{\theta \in \Theta} \mathbb{E}\left[\mathbf{SW}_p(\hat{\mu}_n, \hat{\mu}_{\theta, m(n)}) \big| Y_{1:n}\right] = \inf_{\theta \in \Theta} \mathbf{SW}_p(\mu_\star, \mu_\theta), \quad and \tag{10}$$

$$\limsup_{n \to +\infty} \mathrm{argmin}_{\theta \in \Theta} \mathbb{E}\left[\mathbf{SW}_p(\hat{\mu}_n, \hat{\mu}_{\theta, m(n)}) \big| Y_{1:n}\right] \subset \mathrm{argmin}_{\theta \in \Theta} \mathbf{SW}_p(\mu_\star, \mu_\theta), \tag{11}$$

*where* $\hat{\mu}_n$ *and* $\hat{\mu}_{\theta, m(n)}$ *are defined by* (2) *and* (4) *respectively. Besides, for all* $\omega \in \mathsf{E}$, *there exists* $n(\omega)$ *such that, for all* $n \geq n(\omega)$, *the set* $\mathrm{argmin}_{\theta \in \Theta} \mathbb{E}[\mathbf{SW}_p(\hat{\mu}_n, \hat{\mu}_{\theta, m(n)}) | Y_{1:n}]$ *is non-empty.*

Similar to Theorem 2, this theorem shows that, when the number of observations goes to infinity, the estimator obtained with the expected distance will converge to a global minimizer.

## 3.3 Convergence of MESWE to MSWE

In practical applications, we can only use a finite number of generated samples $Z_{1:m}$. In this subsection, we analyze the case where the observations $Y_{1:n}$ are kept fixed while the number of generated samples increases, *i.e.* $m \to +\infty$ and we show in this scenario that MESWE converges to MSWE, assuming the latter exists.

Before deriving this result, we formulate a technical assumption below.

**A5.** *For some* $\epsilon > 0$ *and* $\epsilon_n = \inf_{\theta \in \Theta} \mathbf{SW}_p(\hat{\mu}_n, \mu_\theta)$, *the set* $\Theta_{\epsilon, n} = \{\theta \in \Theta : \mathbf{SW}_p(\hat{\mu}_n, \mu_\theta) \leq \epsilon_n + \epsilon\}$ *is bounded almost surely.*

**Theorem 4** (MESWE converges to MSWE as $m \to +\infty$). *Assume A1, A4 and A5. Then,*

$$\lim_{m \to +\infty} \inf_{\theta \in \Theta} \mathbb{E}\left[\mathbf{SW}_p(\hat{\mu}_n, \hat{\mu}_{\theta, m}) | Y_{1:n}\right] = \inf_{\theta \in \Theta} \mathbf{SW}_p(\hat{\mu}_n, \mu_\theta) \tag{12}$$

$$\limsup_{m \to +\infty} \mathrm{argmin}_{\theta \in \Theta} \mathbb{E}\left[\mathbf{SW}_p(\hat{\mu}_n, \hat{\mu}_{\theta, m}) | Y_{1:n}\right] \subset \mathrm{argmin}_{\theta \in \Theta} \mathbf{SW}_p(\hat{\mu}_n, \mu_\theta) \tag{13}$$

*Besides, there exists* $m^*$ *such that, for any* $m \geq m^*$, *the set* $\mathrm{argmin}_{\theta \in \Theta} \mathbb{E}\left[\mathbf{SW}_p(\hat{\mu}_n, \hat{\mu}_{\theta, m}) | Y_{1:n}\right]$ *is non-empty.*

This result shows that MESWE would be indeed promising in practice, as one get can more accurate estimations by increasing $m$.

## 3.4 Rate of convergence and the asymptotic distribution

In our last set of theoretical results, we investigate the asymptotic distribution of MSWE and we establish a rate of convergence. We now suppose that we are in the well-specified setting, *i.e.* there exists $\theta_\star$ in the interior of $\Theta$ such that $\mu_{\theta_\star} = \mu_\star$, and we consider the following two assumptions. For any $u \in \mathbb{S}^{d-1}$ and $t \in \mathbb{R}$, we define $F_\theta(u, t) = \int_Y \mathbb{1}_{(-\infty, t]}(\langle u, y\rangle) d\mu_\theta(y)$. Note that for any $u \in \mathbb{S}^{d-1}$, $F_\theta(u, \cdot)$ is the cumulative distribution function (CDF) associated to the measure $u_\sharp^\star \mu_\theta$.

**A6.** *For all $\epsilon > 0$, there exists $\delta > 0$ such that $\inf_{\theta \in \Theta: \rho_\Theta(\theta, \theta_\star) \geq \epsilon} \mathbf{SW}_1(\mu_{\theta_\star}, \mu_\theta) > \delta$ .*

Let $\mathcal{L}^1(\mathbb{S}^{d-1} \times \mathbb{R})$ denote the class of functions that are absolutely integrable on the domain $\mathbb{S}^{d-1} \times \mathbb{R}$, with respect to the measure $d\boldsymbol{\sigma} \otimes$ Leb, where Leb denotes the Lebesgue measure.

**A7.** *Assume that there exists a measurable function $D_\star = (D_{\star,1}, \ldots, D_{\star,d_\theta}) : \mathbb{S}^{d-1} \times \mathbb{R} \mapsto \mathbb{R}^{d_\theta}$ such that for each $i = 1, \ldots, d_\theta$, $D_{\star,i} \in \mathcal{L}^1(\mathbb{S}^{d-1} \times \mathbb{R})$ and*

$$\int_{\mathbb{S}^{d-1}} \int_{\mathbb{R}} |F_\theta(u, t) - F_{\theta_\star}(u, t) - \langle \theta - \theta_\star, D_\star(u, t)\rangle| \, dt d\boldsymbol{\sigma}(u) = \boldsymbol{\epsilon}(\rho_\Theta(\theta, \theta_\star)) \,,$$

*where $\boldsymbol{\epsilon} : \mathbb{R}_+ \to \mathbb{R}_+$ satisfies $\lim_{t \to 0} \boldsymbol{\epsilon}(t) = 0$. Besides, $\{D_{\star,i}\}_{i=1}^{d_\theta}$ are linearly independent in $\mathcal{L}^1(\mathbb{S}^{d-1} \times \mathbb{R})$.*

For any $u \in \mathbb{S}^{d-1}$, and $t \in \mathbb{R}$, define: $\hat{F}_n(u, t) = n^{-1} \operatorname{card}\{i \in \{1, \ldots, n\} : \langle u, Y_i\rangle \leq t\}$, where card denotes the cardinality of a set, and for any $u \in \mathbb{S}^{d-1}$, $\hat{F}_n(u, \cdot)$ is the CDF associated to the measure $u_\sharp^\star \hat{\mu}_n$.

**A8.** *There exists a random element $G_\star : \mathbb{S}^{d-1} \times \mathbb{R} \mapsto \mathbb{R}$ such that the stochastic process $\sqrt{n}(\hat{F}_n - F_{\theta_\star})$ converges weakly in $\mathcal{L}_1(\mathbb{S}^{d-1} \times \mathbb{R})$ to $G_\star$[1].*

**Theorem 5.** *Assume A1, A2, A3, A6, A7 and A8. Then, the asymptotic distribution of the goodness-of-fit statistic is given by,*

$$\sqrt{n} \inf_{\theta \in \Theta} \mathbf{SW}_1(\hat{\mu}_n, \mu_\theta) \xrightarrow{w} \inf_{\theta \in \Theta} \int_{\mathbb{S}^{d-1}} \int_{\mathbb{R}} |G_\star(u, t) - \langle \theta, D_\star(u, t)\rangle| \, dt d\boldsymbol{\sigma}(u), \quad as\ n \to +\infty \,,$$

*where $\hat{\mu}_n$ is defined by* (2).

**Theorem 6.** *Assume A1, A2, A3, A6, A7 and A8. Suppose also that the random map $\theta \mapsto \int_{\mathbb{S}^{d-1}} \int_{\mathbb{R}} |G_\star(u, t) - \langle \theta, D_\star(u, t)\rangle| \, dt d\boldsymbol{\sigma}(u)$ has a unique infimum almost surely. Then, MSWE with $p = 1$ satisfies,*

$$\sqrt{n}(\hat{\theta}_n - \theta_\star) \xrightarrow{w} \operatorname{argmin}_{\theta \in \Theta} \int_{\mathbb{S}^{d-1}} \int_{\mathbb{R}} |G_\star(u, t) - \langle \theta, D_\star(u, t)\rangle| \, dt d\boldsymbol{\sigma}(u), \quad as\ n \to +\infty \,,$$

*where $\hat{\theta}_n$ is defined by* (1) *with $\mathbf{SW}_1$ in place of $\mathbf{D}$.*

These results show that the estimator and the associated goodness-of-fit statistics will converge to a random variable in distribution, where the rate of convergence is $\sqrt{n}$. Note that $G_\star$ is defined as a random element (see **A8**), therefore we can not claim that the convergence in distribution derived in Theorem 5 and 6 implies the convergence in probability.

This CLT is also inspired by [3], where they identified the asymptotic distribution associated to the minimum Wasserstein estimator. However, since $\mathbf{W}_p$ admits an analytical form only when $d = 1$, their result is restricted to the scalar case, and in their conclusion, [3] conjecture that the rate of the minimum Wasserstein estimators would depend negatively on the dimension of the observation space. On the contrary, since $\mathbf{SW}_p$ is defined in terms of one-dimensional $\mathbf{W}_p$ distances, we circumvent the curse of dimensionality and our result holds for any finite dimension. While the perceived computational burden has created a pessimism in the machine learning community about the use of Wasserstein-based methods in large dimensional settings, which motivated the rise of regularized optimal transport [26], we believe that our findings provide an interesting counter-example to this conception.

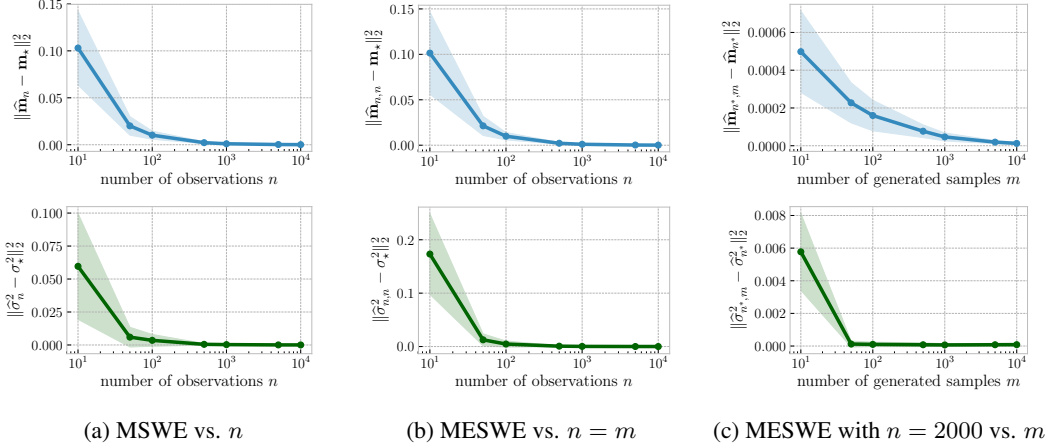

(a) MSWE vs. $n$　　　　(b) MESWE vs. $n = m$　　　　(c) MESWE with $n = 2000$ vs. $m$

Figure 2: Min. SW estimation on Gaussians in $\mathbb{R}^{10}$. Figure 2a and Figure 2b show the mean squared error between $(\mathbf{m}_\star, \sigma_\star^2) = (\mathbf{0}, 1)$ and MSWE $(\hat{\mathbf{m}}_n, \hat{\sigma}_n^2)$ (resp. MESWE $(\hat{\mathbf{m}}_{n,n}, \hat{\sigma}_{n,n}^2)$) for $n$ from 10 to 10 000, illustrating Theorems 2 and 3. Figure 2c shows the error between $(\hat{\mathbf{m}}_n, \hat{\sigma}_n^2)$ and $(\hat{\mathbf{m}}_{n,m}, \hat{\sigma}_{n,m}^2)$ for 2000 observations and $m$ from 10 to 10 000, to illustrate Theorem 4. Results are averaged over 100 runs, the shaded areas represent the standard deviation.

## 4 Experiments

We conduct experiments on synthetic and real data to empirically confirm our theorems. We explain in Section 4 of the supplementary document the optimization methods used to find the estimators. Specifically, we can use stochastic iterative optimization algorithm (e.g., stochastic gradient descent). Note that, since we calculate (expected) SW with Monte Carlo approximations over a finite set of projections (and a finite number of *'generated datasets'*), MSWE and MESWE fall into the category of doubly stochastic algorithms. Our experiments on synthetic data actually show that using only one random projection and one randomly generated dataset at each iteration of the optimization process is enough to illustrate our theorems. We provide the code to reproduce the experiments.[2]

**Multivariate Gaussian distributions:** We consider the task of estimating the parameters of a 10-dimensional Gaussian distribution using our SW estimators: we are interested in the model $\mathcal{M} = \{\mathcal{N}(\mathbf{m}, \sigma^2 \mathbf{I}) : \mathbf{m} \in \mathbb{R}^{10}, \sigma^2 > 0\}$ and we draw i.i.d. observations with $(\mathbf{m}_\star, \sigma_\star^2) = (\mathbf{0}, 1)$. The advantage of this simple setting is that the density of the generated data has a closed-form expression, which makes MSWE tractable. We empirically verify our central limit theorem: for different values of $n$, we compute 500 times MSWE of order 1 using one random projection, then we estimate the density of $\hat{\sigma}_n^2$ with a kernel density estimator. Figure 1 shows the distributions centered and rescaled by $\sqrt{n}$ for each $n$, and confirms the convergence rate

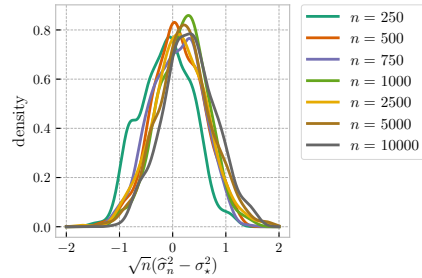

Figure 1: Probability density estimates of the MSWE $\hat{\sigma}_n^2$ of order 1, centered and rescaled by $\sqrt{n}$, on the 10-dimensional Gaussian model for different values of $n$.

that we derived (Theorem 6). To illustrate the consistency property in Theorem 2, we approximate MSWE of order 2 for different numbers of observed data $n$ using one random projection and we report for each $n$ the mean squared error between the estimate mean and variance and the data-generating parameters $(\mathbf{m}_\star, \sigma_\star^2)$. We proceed the same way to study the consistency of MESWE (Theorem 3), which we approximate using one random projections and one generated dataset $z_{1:m}$ of size $m = n$ for different values of $n$. We also verify the convergence of MESWE to MSWE (Theorem 4): we compute these estimators on a fixed set of $n = 2000$ observations for different $m$, and we measure

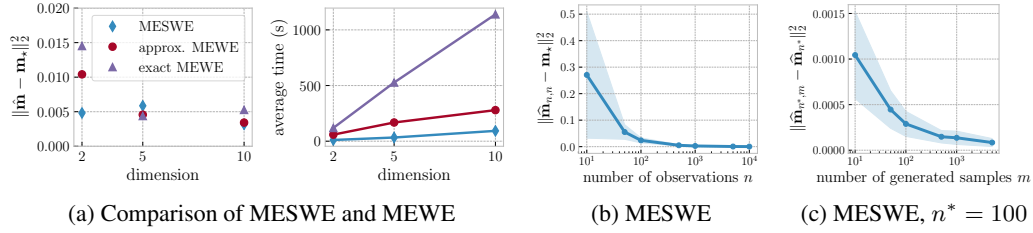

(a) Comparison of MESWE and MEWE      (b) MESWE      (c) MESWE, $n^* = 100$

Figure 3: Min. SW estimation for the location parameter of multivariate elliptically contoured stable distributions. Figure 3a compares the quality of the estimation provided by SW and Wasserstein-based estimators as well as their average computational time, for different values of dimension $d$. Figure 3b and Figure 3c illustrate, for $d = 10$, the consistency of MESWE $\hat{\mathbf{m}}_{n,m}$ and its convergence to the MSWE $\hat{\mathbf{m}}_n$. Results are averaged over 100 runs, the shaded area represent the standard deviation.

the error between them for each $m$. Results are shown in Figure 2. We see that our estimators indeed converge to $(\mathbf{m}_\star, \sigma_\star^2)$ as the number of observations increases (Figures 2a, 2b), and on a fixed observed dataset, MESWE converges to MSWE as we generate more samples (Figure 2c).

**Multivariate elliptically contoured stable distributions:** We focus on parameter inference for a subclass of multivariate stable distributions, called elliptically contoured stable distributions and denoted by $\mathcal{E}\alpha\mathcal{S}_c$ [27]. Stable distributions refer to a family of heavy-tailed probability distributions that generalize Gaussian laws and appear as the limit distributions in the generalized central limit theorem [28]. These distributions have many attractive theoretical properties and have been proven useful in modeling financial [29] data or audio signals [30, 31]. While special univariate cases include Gaussian, Lévy and Cauchy distributions, the density of stable distributions has no general analytic form, which restricts their practical application, especially for the multivariate case.

If $Y \in \mathbb{R}^d \sim \mathcal{E}\alpha\mathcal{S}_c(\mathbf{\Sigma}, \mathbf{m})$, then its joint characteristic function is defined for any $\mathbf{t} \in \mathbb{R}^d$ as $\mathbb{E}[\exp(i\mathbf{t}^T Y)] = \exp\left(-(\mathbf{t}^T\mathbf{\Sigma}\mathbf{t})^{\alpha/2} + i\mathbf{t}^T\mathbf{m}\right)$, where $\mathbf{\Sigma}$ is a positive definite matrix (akin to a correlation matrix), $\mathbf{m} \in \mathbb{R}^d$ is a location vector (equal to the mean if it exists) and $\alpha \in (0, 2)$ controls the thickness of the tail. Even though their densities cannot be evaluated easily, it is straightforward to sample from $\mathcal{E}\alpha\mathcal{S}_c$ [27], therefore it is particularly relevant here to apply MESWE instead of MLE.

To demonstrate the computational advantage of MESWE over the minimum expected Wasserstein estimator [3, MEWE], we consider observations in $\mathbb{R}^d$ i.i.d. from $\mathcal{E}\alpha\mathcal{S}_c(\mathbf{I}, \mathbf{m}_\star)$ where each component of $\mathbf{m}_\star$ is 2 and $\alpha = 1.8$, and $\mathcal{M} = \left\{\mathcal{E}\alpha\mathcal{S}_c(\mathbf{I}, \mathbf{m}) : \mathbf{m} \in \mathbb{R}^d\right\}$. The Wasserstein distance on multivariate data is either computed exactly by solving the linear program in (5), or approximated by solving a regularized version of this problem with Sinkhorn's algorithm [12]. The MESWE is approximated using 10 random projections and 10 sets of generated samples. Then, following the approach in [3], we use the gradient-free optimization method Nelder-Mead to minimize the Wasserstein and SW distances. We report on Figure 3a the mean squared error between each estimate and $\mathbf{m}_\star$, as well as their average computational time for different values of dimension $d$. We see that MESWE provides the same quality of estimation as its Wasserstein-based counterparts while considerably reducing the computational time, especially in higher dimensions. We focus on this model in $\mathbb{R}^{10}$ and we illustrate the consistency of the MESWE $\hat{\mathbf{m}}_{n,m}$, approximated with one random projection and one generated dataset, the same way as for the Gaussian model: see Figure 3b. To confirm the convergence of $\hat{\mathbf{m}}_{n,m}$ to the MSWE $\hat{\mathbf{m}}_n$, we fix $n = 100$ observations and we compute the mean squared error between the two approximate estimators (using one random projection and one generated dataset) for different values of $m$ (Figure 3c). Note that the MSWE is approximated with the MESWE obtained for a large enough value of $m$: $\hat{\mathbf{m}}_n \approx \hat{\mathbf{m}}_{n,10\,000}$.

**High-dimensional real data using GANs:** Finally, we run experiments on image generation using the Sliced-Wasserstein Generator (SWG), an alternative GAN formulation based on the minimization of the SW distance [16]. Specifically, the generative modeling approach consists in introducing a random variable $Z$ which takes value in $\mathsf{Z}$ with a fixed distribution, and then transforming $Z$ through a neural network. This defines a parametric function $T_\theta : \mathsf{Z} \to \mathsf{Y}$ that is able to produce images from a distribution $\mu_\theta$.

The goal is to optimize the neural network parameters such that the generated images are close to the observed ones. [16] proposes to minimize the SW distance between $\mu_\theta$ and the real data distribution over $\theta$ as the generator objective, and train on MESWE in practice. For our experiments, we design a neural network with the fully-connected configuration given in [16, Appendix D] and we use the MNIST dataset, made of $60\,000$ training images and $10\,000$ test images of size $28 \times 28$. Our training objective is MESWE of order 2 approximated with 20 random projections and 20 different generated datasets. We

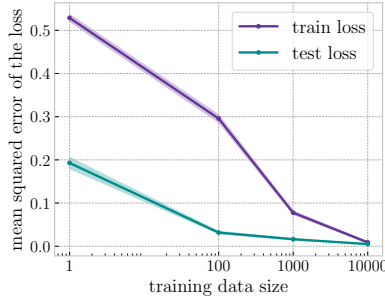

Figure 4: Mean-squared error between the training (test) loss for $(n, m) \in \big\{ (1,1), (100, 20), (1000, 40), (10\,000, 60) \big\}$ and the training (test) loss for $(n, m) = (60\,000, 200)$ on MNIST using the SW generator. We trained for $20\,000$ iterations with the ADAM optimizer [32].

study the consistent behavior of the MESWE by training the neural network on different sizes $n$ of training data and different numbers $m$ of generated samples and by comparing the final training loss and test loss to the ones obtained when learning on the whole training dataset ($n = 60\,000$) and $m = 200$. Results are averaged over 10 runs and shown on Figure 4, where the shaded areas correspond to the standard deviation over the runs. We observe that our results confirm Theorem 3.

We would like to point out that, in all of our experiments, the random projections used in the Monte Carlo average that estimates the integral in (7) were picked uniformly on $\mathbb{S}^{d-1}$ (see Section 4 in the supplementary document for more details). The sampling on $\mathbb{S}^{d-1}$ directly impacts the quality of the resulting approximation of SW, and might induce variance in practice when learning generative models. On the theoretical side, studying the asymptotic properties of SW-based estimators obtained with a finite number of projections is an interesting question (e.g., their behavior might depend on the sampling method or the number of projections used). We leave this study for future research.

## 5 Conclusion

The Sliced-Wasserstein distance has been an attractive metric choice for learning in generative models, where the densities cannot be computed directly. In this study, we investigated the asymptotic properties of estimators that are obtained by minimizing SW and the expected SW. We showed that (i) convergence in SW implies weak convergence of probability measures in general Wasserstein spaces, (ii) the estimators are consistent, (iii) the estimators converge to a random variable in distribution with a rate of $\sqrt{n}$. We validated our mathematical results on both synthetic data and neural networks. We believe that our techniques can be further extended to the extensions of SW such as [20, 33, 34].

**Acknowledgements**

The authors are grateful to Pierre Jacob for his valuable comments on an earlier version of this manuscript. This work is partly supported by the French National Research Agency (ANR) as a part of the FBIMATRIX project (ANR-16-CE23-0014) and by the industrial chair Machine Learning for Big Data from Télécom ParisTech. Alain Durmus acknowledges support from Polish National Science Center grant: NCN UMO-2018/31/B/ST1/00253.

## Footnotes

[1]Under mild assumptions on the tails of $u_\sharp^\star \mu_\star$ for any $u \in \mathbb{S}^{d-1}$, we believe that one can prove that **A8** holds in general by extending [24, Proposition 3.5] and [25, Theorem 2.1a].

[2]See https://github.com/kimiandj/min_swe.

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
