[Supplementary Material]

# Asymptotic Guarantees for Learning Generative Models with the Sliced-Wasserstein Distance
## SUPPLEMENTARY DOCUMENT

**Kimia Nadjahi**[1], **Alain Durmus**[2], **Umut Şimşekli**[1,3], **Roland Badeau**[1]
1: LTCI, Télécom Paris, Institut Polytechnique de Paris, France
2: CMLA, ENS Cachan, CNRS, Université Paris-Saclay, France
3: Department of Statistics, University of Oxford, UK
{kimia.nadjahi, umut.simsekli, roland.badeau}@telecom-paris.fr
alain.durmus@cmla.ens-cachan.fr

## 1 Preliminaries

### 1.1 Convergence and lower semi-continuity

**Definition 1** (Weak convergence). *Let $(\mu_k)_{k\in\mathbb{N}}$ be a sequence of probability measures on $\mathsf{Y}$. We say that $\mu_k$ converges weakly to a probability measure $\mu$ on $\mathsf{Y}$, and write $(\mu_k)_{k\in\mathbb{N}} \xrightarrow{w} \mu$ (or $\mu_k \xrightarrow{w} \mu$), if for any continous and bounded function $f : \mathsf{Y} \to \mathbb{R}$, we have*

$$\lim_{k\to+\infty} \int f \, \mathrm{d}\mu_k = \int f \, \mathrm{d}\mu .$$

**Definition 2** (Epi-convergence). *Let $\Theta$ be a metric space and $f : \Theta \to \mathbb{R}$. Consider a sequence $(f_k)_{k\in\mathbb{N}}$ of functions from $\Theta$ to $\mathbb{R}$. We say that the sequence $(f_k)_{k\in\mathbb{N}}$ epi-converges to a function $f : \Theta \to \mathbb{R}$, and write $(f_k)_{k\in\mathbb{N}} \xrightarrow{e} f$, if for each $\theta \in \Theta$,*

$$\liminf_{k\to\infty} f_k(\theta_k) \geq f(\theta) \text{ for every sequence } (\theta_k)_{n\in\mathbb{N}} \text{ such that } \lim_{k\to+\infty} \theta_k = \theta ,$$

$$\text{and} \quad \limsup_{k\to\infty} f_k(\theta_k) \leq f(\theta) \text{ for a sequence } (\theta_k)_{n\in\mathbb{N}} \text{ such that } \lim_{k\to+\infty} \theta_k = \theta .$$

An equivalent and useful characterization of epi-convergence is given in [1, Proposition 7.29], which we paraphrase in Proposition S4 after recalling the definition of lower semi-continuous functions.

**Definition 3** (Lower semi-continuity). *Let $\Theta$ be a metric space and $f : \Theta \to \mathbb{R}$. We say that $f$ is lower semi-continuous (l.s.c.) on $\Theta$ if for any $\theta_0 \in \Theta$,*

$$\liminf_{\theta\to\theta_0} f(\theta) \geq f(\theta_0)$$

**Proposition S4** (Characterization of epi-convergence via minimization, Proposition 7.29 of [1]). *Let $\Theta$ be a metric space and $f : \Theta \to \mathbb{R}$ be a l.s.c. function. The sequence $(f_k)_{k\in\mathbb{N}}$, with $f_k : \Theta \to \mathbb{R}$ for any $n \in \mathbb{N}$, epi-converges to $f$ if and only if*

*(a)* $\liminf_{k\to\infty} \inf_{\theta\in\mathsf{K}} f_k(\theta) \geq \inf_{\theta\in\mathsf{K}} f(\theta)$ *for every compact set* $\mathsf{K} \subset \Theta$ ;

*(b)* $\limsup_{k\to\infty} \inf_{\theta\in\mathsf{O}} f_k(\theta) \leq \inf_{\theta\in\mathsf{O}} f(\theta)$ *for every open set* $\mathsf{O} \subset \Theta$.

[1, Theorem 7.31], paraphrased below, gives asymptotic properties for the infimum and argmin of epiconvergent functions and will be useful to prove the existence and consistency of our estimators.

**Theorem S5** (Inf and argmin in epiconvergence, Theorem 7.31 of [1]). *Let $\Theta$ be a metric space, $f : \Theta \to \mathbb{R}$ be a l.s.c. function and $(f_k)_{k\in\mathbb{N}}$ be a sequence with $f_k : \Theta \to \mathbb{R}$ for any $n \in \mathbb{N}$. Suppose $(f_k)_{k\in\mathbb{N}} \xrightarrow{e} f$ with $-\infty < \inf_{\theta\in\Theta} f(\theta) < \infty$.*

*(a)* *It holds* $\lim_{k\to\infty} \inf_{\theta\in\Theta} f_k(\theta) = \inf_{\theta\in\Theta} f(\theta)$ *if and only if for every* $\eta > 0$ *there exists a compact set* $\mathsf{K} \subset \Theta$ *and* $N \in \mathbb{N}$ *such for any* $k \geq N$,

$$\inf_{\theta\in\mathsf{K}} f_k(\theta) \leq \inf_{\theta\in\Theta} f_k(\theta) + \eta .$$

*(b)* *In addition,* $\limsup_{k\to\infty} \operatorname{argmin}_{\theta\in\Theta} f_k(\theta) \subset \operatorname{argmin}_{\theta\in\Theta} f(\theta)$.

## 2 Preliminary results

In this section, we gather technical results regarding lower semi-continuity of (expected) Sliced-Wasserstein distances and measurability of MSWE which will be needed in our proofs.

### 2.1 Lower semi-continuity of Sliced-Wasserstein distances

**Lemma S6** (Lower semi-continuity of $\mathbf{SW}_p$)**.** *Let* $p \in [1, \infty)$. *The Sliced-Wasserstein distance of order* $p$ *is lower semi-continuous on* $\mathcal{P}_p(\mathsf{Y}) \times \mathcal{P}_p(\mathsf{Y})$ *endowed with the topology of weak convergence, i.e. for any sequences* $(\mu_k)_{k\in\mathbb{N}}$ *and* $(\nu_k)_{k\in\mathbb{N}}$ *of* $\mathcal{P}_p(\mathsf{Y})$ *which converge weakly to* $\mu \in \mathcal{P}_p(\mathsf{Y})$ *and* $\nu \in \mathcal{P}_p(\mathsf{Y})$ *respectively, we have:*

$$\mathbf{SW}_p(\mu, \nu) \leq \liminf_{k\to+\infty} \mathbf{SW}_p(\mu_k, \nu_k) .$$

*Proof.* First, by the continuous mapping theorem, if a sequence $(\mu_k)_{k\in\mathbb{N}}$ of elements of $\mathcal{P}_p(\mathsf{Y})$ converges weakly to $\mu$, then for any continuous function $f : \mathsf{Y} \to \mathbb{R}$, $(f_\sharp \mu_k)_{k\in\mathbb{N}}$ converges weakly to $f_\sharp \mu$. In particular, for any $u \in \mathbb{S}^{d-1}$, $u_\sharp^\star \mu_k \xrightarrow{w} u_\sharp^\star \mu$ since $u^\star$ is a bounded linear form thus continuous.

Let $p \in [1, \infty)$. We introduce the two sequences $(\mu_k)_{k\in\mathbb{N}}$ and $(\nu_k)_{k\in\mathbb{N}}$ of elements of $\mathcal{P}_p(\mathsf{Y})$ such that $\mu_k \xrightarrow{w} \mu$ and $\nu_k \xrightarrow{w} \nu$. We show that for any $u \in \mathbb{S}^{d-1}$,

$$\mathbf{W}_p^p(u_\sharp^\star \mu, u_\sharp^\star \nu) \leq \liminf_{k\to+\infty} \mathbf{W}_p^p(u_\sharp^\star \mu_k, u_\sharp^\star \nu_k) . \tag{S1}$$

Indeed, if (S1) holds, then the proof is completed using the definition of the Sliced-Wasserstein distance (7) and Fatou's Lemma. Let $u \in \mathbb{S}^{d-1}$. For any $k \in \mathbb{N}$, let $\gamma_k \in \mathcal{P}(\mathbb{R} \times \mathbb{R})$ be an optimal transference plan between $u_\sharp^\star \mu_k$ and $u_\sharp^\star \nu_k$ for the Wasserstein distance of order $p$ which exists by [2, Theorem 4.1] *i.e.*

$$\mathbf{W}_p^p(u_\sharp^\star \mu_k, u_\sharp^\star \nu_k) = \int_{\mathbb{R}\times\mathbb{R}} |a - b|\, \mathrm{d}\gamma_k(a, b) .$$

Note that by [2, Lemma 4.4] and Prokhorov's Theorem, $(\gamma_k)_{k\in\mathbb{N}}$ is sequentially compact in $\mathcal{P}(\mathbb{R} \times \mathbb{R})$ for the topology associated with the weak convergence. Now, consider a subsequence $(\gamma_{\phi_1(k)})_{k\in\mathbb{N}}$ where $\phi_1 : \mathbb{N} \to \mathbb{N}$ is increasing such that

$$\lim_{k\to+\infty} \int_{\mathbb{R}\times\mathbb{R}} |a - b|^p\, \mathrm{d}\gamma_{\phi_1(k)}(a, b) = \lim_{k\to+\infty} \mathbf{W}_p^p(u_\sharp^\star \mu_{\phi_1(k)}, u_\sharp^\star \nu_{\phi_1(k)})$$
$$= \liminf_{k\to+\infty} \mathbf{W}_p^p(u_\sharp^\star \mu_k, u_\sharp^\star \nu_k) . \tag{S2}$$

Since $(\gamma_k)_{k\in\mathbb{N}}$ is sequentially compact, $(\gamma_{\phi_1(k)})_{k\in\mathbb{N}}$ is sequentially compact as well, and therefore there exists an increasing function $\phi_2 : \mathbb{N} \to \mathbb{N}$ and a probability distribution $\gamma \in \mathcal{P}(\mathbb{R} \times \mathbb{R})$ such that $(\gamma_{\phi_2(\phi_1(k))})_{k\in\mathbb{N}}$ converges weakly to $\gamma$. Then, we obtain by (S2),

$$\int_{\mathbb{R}\times\mathbb{R}} \|a - b\|^p\, \mathrm{d}\gamma(a, b) = \lim_{k\to+\infty} \int_{\mathbb{R}\times\mathbb{R}} \|a - b\|^p\, \mathrm{d}\gamma_{\phi_2(\phi_1(k))}(a, b) = \liminf_{k\to+\infty} \mathbf{W}_p^p(u_\sharp^\star \mu_k, u_\sharp^\star \nu_k) .$$

If we show that $\gamma \in \Gamma(u_\sharp^\star \mu, u_\sharp^\star \nu)$, it will conclude the proof of (S1) by definition of the Wasserstein distance (5). But for any continuous and bounded function $f : \mathbb{R} \to \mathbb{R}$, since for any $k \in \mathbb{N}$, $\gamma_k \in \Gamma(\mu_k, \nu_k)$, and $(\mu_k)_{k\in\mathbb{N}}, (\nu_k)_{k\in\mathbb{N}}$ converge weakly to $\mu$ and $\nu$ respectively, we have:

$$\int_{\mathbb{R}\times\mathbb{R}} f(a)\mathrm{d}\gamma(a, b) = \lim_{k\to+\infty} \int_{\mathbb{R}\times\mathbb{R}} f(a)\mathrm{d}\gamma_{\phi_2(\phi_1(k))}(a, b) = \lim_{k\to+\infty} \int_{\mathbb{R}} f(a)\mathrm{d}u_\sharp^\star \mu_{\phi_2(\phi_1(k))}(a)$$
$$= \int_{\mathbb{R}} f(a)\mathrm{d}u_\sharp^\star \mu(a) ,$$

and similarly

$$\int_{\mathbb{R}\times\mathbb{R}} f(b)\mathrm{d}\gamma(a,b) = \int_{\mathbb{R}} f(b)\mathrm{d}u_{\sharp}^{\star}\nu(a) \ .$$

This shows that $\gamma \in \Gamma(u_{\sharp}^{\star}\mu, u_{\sharp}^{\star}\nu)$ and therefore, (S1) is true. We conclude by applying Fatou's Lemma.

$\square$

By a direct application of Lemma S6, we have the following result.

**Corollary 7.** *Assume A1. Then, $(\mu,\theta) \mapsto \mathbf{SW}_p(\mu,\mu_\theta)$ is lower semi-continuous in $\mathcal{P}_p(\mathsf{Y}) \times \Theta$.*

**Lemma S8** (Lower semi-continuity of $\mathbb{E}\mathbf{SW}_p$). *Let $p \in [1,\infty)$ and $m \in \mathbb{N}^*$. Denote for any $\mu \in \mathcal{P}_p(\mathsf{Y})$, $\hat{\mu}_m = (1/m)\sum_{i=1}^{m}\delta_{Z_i}$, where $Z_{1:m}$ are i.i.d. samples from $\mu$. Then, the map $(\nu,\mu) \mapsto \mathbb{E}\left[\mathbf{SW}_p(\nu,\hat{\mu}_m)\right]$ is lower semi-continuous on $\mathcal{P}_p(\mathsf{Y}) \times \mathcal{P}_p(\mathsf{Y})$ endowed with the topology of weak convergence.*

*Proof.* We consider two sequences $(\mu_k)_{k\in\mathbb{N}}$ and $(\nu_k)_{k\in\mathbb{N}}$ of probability measures in $\mathsf{Y}$, such that $(\mu_k)_{k\in\mathbb{N}} \xrightarrow{w} \mu$ and $(\nu_k)_{k\in\mathbb{N}} \xrightarrow{w} \nu$, and we fix $m \in \mathbb{N}^*$.

By Skorokhod's representation theorem, there exists a probability space $(\tilde{\Omega},\tilde{\mathcal{F}},\tilde{\mathbb{P}})$, a sequence of random variables $(\tilde{X}_k^1,\ldots,\tilde{X}_k^m)_{k\in\mathbb{N}}$ and a random variable $(\tilde{X}^1,\ldots,\tilde{X}^m)$ defined on $\tilde{\Omega}$ such that for any $k \in \mathbb{N}$ and $i \in \{1,\ldots,m\}$, $\tilde{X}_k^i$ has distribution $\mu_k$, $\tilde{X}^i$ has distribution $\mu$ and $(\tilde{X}_k^1,\ldots,\tilde{X}_k^m)_{k\in\mathbb{N}^*}$ converges to $(\tilde{X}^1,\ldots,\tilde{X}^m)$, $\tilde{\mathbb{P}}$-almost surely. We then show that the sequence of (random) empirical distributions $(\hat{\mu}_{k,m})_{k\in\mathbb{N}}$ defined by $\hat{\mu}_{k,m} = (1/m)\sum_{i=1}^{m}\delta_{\tilde{X}_k^i}$, weakly converges to $\hat{\mu}_m = (1/m)\sum_{i=1}^{m}\delta_{\tilde{X}^i}$, $\tilde{\mathbb{P}}$-almost surely. Note that it is sufficient to show that for any deterministic sequence $(x_k^1,\ldots,x_k^m)_{k\in\mathbb{N}^*}$ which converges to $(x^1,\ldots,x^m)$, *i.e.* $\lim_{k\to+\infty}\max_{i\in\{1,\ldots,m\}}\rho(x_k^i,x^i) = 0$, then the sequence of empirical distributions $(\hat{\nu}_{k,m})_{k\in\mathbb{N}}$ defined by $\hat{\nu}_{k,m} = (1/m)\sum_{i=1}^{m}\delta_{x_k^i}$, weakly converges to $\hat{\nu}_m = (1/m)\sum_{i=1}^{m}\delta_{x^i}$. Note that since the Lévy-Prokhorov metric $\mathbf{d}_{\mathcal{P}}$ metrizes the weak convergence by [3, Theorem 6.8], we only need to show that $\lim_{k\to+\infty}\mathbf{d}_{\mathcal{P}}(\hat{\nu}_{k,m},\hat{\nu}_m) = 0$. More precisely, since for any probability measure $\zeta_1$ and $\zeta_2$,

$$\mathbf{d}_{\mathcal{P}}(\zeta_1,\zeta_2) = \inf\left\{\epsilon > 0 \ : \ \text{for any } \mathsf{A} \in \mathcal{Y}, \ \zeta_1(\mathsf{A}) \le \zeta_2(\mathsf{A}^\epsilon) + \epsilon \text{ and } \zeta_2(\mathsf{A}) \le \zeta_1(\mathsf{A}^\epsilon) + \epsilon\right\} \ ,$$

where $\mathcal{Y}$ is the Borel $\sigma$-field of $(\mathsf{Y},\rho)$ and for any $\mathsf{A} \in \mathcal{Y}$, $\mathsf{A}^\epsilon = \{x \in \mathsf{Y} \ : \ \rho(x,y) < \epsilon \text{ for any } y \in \mathsf{A}\}$, we get

$$\mathbf{d}_{\mathcal{P}}(\hat{\nu}_{k,m},\hat{\nu}_m) \le 2\max_{i\in\{1,\ldots,m\}}\rho(x_k^i,x^i) \ ,$$

and therefore $\lim_{k\to+\infty}\mathbf{d}_{\mathcal{P}}(\hat{\nu}_{k,m},\hat{\nu}_m) = 0$, so that, $(\hat{\nu}_{k,m})_{k\in\mathbb{N}}$ weakly converges to $\hat{\nu}_m$.

Finally, we have that $\hat{\mu}_{k,m} = (1/m)\sum_{i=1}^{m}\delta_{\tilde{X}_k^i}$, weakly converges to $\hat{\mu}_m = (1/m)\sum_{i=1}^{m}\delta_{\tilde{X}^i}$, $\tilde{\mathbb{P}}$-almost surely and we obtain the final result using the lower semi-continuity of the Sliced-Wasserstein distance derived in Lemma S6 and Fatou's lemma which give

$$\tilde{\mathbb{E}}\left[\mathbf{SW}_p(\nu,\hat{\mu}_m)\right] \le \tilde{\mathbb{E}}\left[\liminf_{i\to\infty}\mathbf{SW}_p(\nu_i,\hat{\mu}_{m,i})\right] \le \liminf_{i\to\infty}\tilde{\mathbb{E}}\left[\{\mathbf{SW}_p(\nu_i,\hat{\mu}_{m,i})\right] \ ,$$

where $\tilde{\mathbb{E}}$ is the expectation corresponding to $\tilde{\mathbb{P}}$.

$\square$

The following corollary is a direct consequence of Lemma S8.

**Corollary 9.** *Assume A1. Then, $(\nu,\theta) \mapsto \mathbb{E}[\mathbf{SW}_p(\nu,\hat{\mu}_{\theta,m})|Y_{1:n}]$ is lower semi-continuous on $\mathcal{P}(\mathsf{Y}) \times \Theta$.*

## 2.2 Measurability of the MSWE and MESWE

The measurability of the MSWE and MESWE follows from the application of [4, Corollary 1], also used in [5] and [6], and which we recall in Theorem S10.

**Theorem S10** (Corollary 1 in [4]). *Let* $\mathsf{U}, \mathsf{V}$ *be Polish spaces and $f$ be a real-valued Borel measurable function defined on a Borel subset $\mathsf{D}$ of $\mathsf{U} \times \mathsf{V}$. We denote by* $\mathrm{proj}(\mathsf{D})$ *the set defined as*

$$\mathrm{proj}(\mathsf{D}) = \{u \ : \ there \ exists \ v \in \mathsf{V}, \ (u,v) \in \mathsf{D}\} .$$

*Suppose that for each $u \in \mathrm{proj}(\mathsf{D})$, the section $\mathsf{D}_u = \{v \in V, (u,v) \in \mathsf{D}\}$ is $\sigma$-compact and $f(u, \cdot)$ is lower semi-continuous with respect to the relative topology on $\mathsf{D}_u$. Then,*

1. *The sets* $\mathrm{proj}(\mathsf{D})$ *and* $\mathsf{I} = \{u \in \mathrm{proj}(\mathsf{D}), \ for \ some \ v \in \mathsf{D}_u, \ f(u,v) = \inf f_u\}$ *are Borel*

2. *For each $\epsilon > 0$, there is a Borel measurable function $\phi_\epsilon$ satisfying, for $u \in \mathrm{proj}(\mathsf{D})$,*

$$
\begin{aligned}
f(u, \phi_\epsilon(u)) &= \inf_{\mathsf{D}_u} f_u, & if \ \ u \in \mathsf{I}, \\
&\leq \epsilon + \inf_{\mathsf{D}_u} f_u, & if \ \ u \notin \mathsf{I}, \ \ and \ \ \inf_{\mathsf{D}_u} f_u \neq -\infty \\
&\leq -\epsilon^{-1}, & if \ \ u \notin \mathsf{I}, \ \ and \ \ \inf_{\mathsf{D}_u} f_u = -\infty .
\end{aligned}
$$

**Theorem S11** (Measurability of the MSWE). *Assume A1. For any $n \geq 1$ and $\epsilon > 0$, there exists a Borel measurable function $\hat{\theta}_{n,\epsilon} : \Omega \to \Theta$ that satisfies: for any $\omega \in \Omega$,*

$$
\hat{\theta}_{n,\epsilon}(\omega) \in \left\{
\begin{array}{ll}
\mathrm{argmin}_{\theta \in \Theta} \ \mathbf{SW}_p(\hat{\mu}_n(\omega), \mu_\theta), & if \ this \ set \ is \ non-empty, \\
\{\theta \in \Theta \ : \ \mathbf{SW}_p(\hat{\mu}_n(\omega), \mu_\theta) \leq \epsilon_\star + \epsilon\}, & otherwise.
\end{array}
\right.
$$

*where $\epsilon_\star = \inf_{\theta \in \Theta} \mathbf{SW}_p(\mu_\star, \mu_\theta)$.*

*Proof.* The proof consists in showing that the conditions of Theorem S10 are satisfied.

The empirical measure $\hat{\mu}_n(\omega)$ depends on $\omega \in \Omega$ only through $y = (y_1, \ldots, y_n) \in \mathsf{Y}^n$, so we can consider it as a function on $\mathsf{Y}^n$ rather than on $\Omega$. We introduce $\mathsf{D} = \mathsf{Y}^n \times \Theta$. Since $\mathsf{Y}$ is Polish, $\mathsf{Y}^n$ ($n \in \mathbb{N}^*$) endowed with the product topology is Polish. For any $y \in \mathsf{Y}^n$, the set $\mathsf{D}_y = \{\theta \in \Theta, (y, \theta) \in \mathsf{D}\} = \Theta$ is assumed to be $\sigma$-compact.

The map $y \mapsto \hat{\mu}_n(y)$ is continuous for the weak topology (see the proof of Lemma S8), as well as the map $\theta \mapsto \mu_\theta$ according to **A1**. We deduce by Corollary 7 that the map $(\mu, \theta) \mapsto \mathbf{SW}_p(\mu, \mu_\theta)$ is l.s.c. for the weak topology. Since the composition of a lower semi-continuous function with a continuous function is l.s.c. , the map $(y, \theta) \mapsto \mathbf{SW}_p(\hat{\mu}_n(y), \mu_\theta)$ is l.s.c. for the weak topology, thus measurable and for any $y \in \mathsf{Y}^n$, $\theta \mapsto \mathbf{SW}_p(\hat{\mu}_n(y), \mu_\theta)$ is l.s.c. on $\Theta$. A direct application of Theorem S10 finalizes the proof.

$\square$

**Theorem S12** (Measurability of the MESWE). *Assume A1. For any $n \geq 1$, $m \geq 1$ and $\epsilon > 0$, there exists a Borel measurable function $\hat{\theta}_{n,m,\epsilon} : \Omega \to \Theta$ that satisfies: for any $\omega \in \Omega$,*

$$
\hat{\theta}_{n,m,\epsilon}(\omega) \in \left\{
\begin{array}{ll}
\mathrm{argmin}_{\theta \in \Theta} \ \mathbb{E}\left[\mathbf{SW}_p(\hat{\mu}_n(\omega), \hat{\mu}_{\theta,m})|Y_{1:n}\right], & if \ this \ set \ is \ non-empty, \\
\{\theta \in \Theta \ : \ \mathbb{E}\left[\mathbf{SW}_p(\hat{\mu}_n(\omega), \hat{\mu}_{\theta,m})|Y_{1:n}\right] \leq \epsilon_* + \epsilon\}, & otherwise.
\end{array}
\right.
$$

*where $\epsilon_* = \inf_{\theta \in \Theta} \mathbb{E}[\mathbf{SW}_p(\hat{\mu}_n(\omega), \hat{\mu}_{\theta,m})|Y_{1:n}]$.*

*Proof.* The proof can be done similarly to the proof of Theorem S11: we verify that we can apply Theorem S10 using Corollary 9 instead of Corollary 7. $\square$

## 3 Postponed proofs

### 3.1 Proof of Theorem 1

**Lemma S13.** *Let $(\mu_k)_{k \in \mathbb{N}}$ be a sequence of probability measures on $\mathbb{R}^d$ and $\mu$ a measure in $\mathbb{R}^d$ such that,*

$$\lim_{k \to \infty} \mathbf{SW}_1(\mu_k, \mu) = 0 .$$

*Then, there exists an increasing function $\phi : \mathbb{N} \to \mathbb{N}$ such that the subsequence $(\mu_{\phi(k)})_{k \in \mathbb{N}}$ converges weakly to $\mu$.*

*Proof.* By definition, we have that

$$\lim_{k\to\infty} \int_{\mathbb{S}^{d-1}} \mathbf{W}_1(u^\star_\sharp \mu_k, u^\star_\sharp \mu) \mathrm{d}\boldsymbol{\sigma}(u) = 0 \ .$$

Therefore by [7, Theorem 2.2.5], for $\boldsymbol{\sigma}$-almost every ($\boldsymbol{\sigma}$-a.e.) $u \in \mathbb{S}^{d-1}$, there exists a subsequence $(u^\star_\sharp \mu_{\phi(k)})_{k\in\mathbb{N}}$ with $\phi : \mathbb{N} \to \mathbb{N}$ increasing, such that $\lim_{k\to\infty} \mathbf{W}_1(u^\star_\sharp \mu_{\phi(k)}, u^\star_\sharp \mu) = 0$ . By [2, Theorem 6.9], it implies that for $\boldsymbol{\sigma}$-a.e. $u \in \mathbb{S}^{d-1}$, $(u^\star_\sharp \mu_{\phi(k)})_{k\in\mathbb{N}} \xrightarrow{w} u^\star_\sharp \mu$. Lévy's characterization [8, Theorem 4.3] gives that, for $\boldsymbol{\sigma}$-a.e. $u \in \mathbb{S}^{d-1}$ and any $s \in \mathbb{R}$,

$$\lim_{k\to\infty} \Phi_{u^\star_\sharp \mu_{\phi(k)}}(s) = \Phi_{u^\star_\sharp \mu}(s) \ ,$$

where, for any distribution $\nu \in \mathcal{P}(\mathbb{R}^p)$, $\Phi_\nu$ denotes the characteristic function of $\nu$ and is defined for any $v \in \mathbb{R}^p$ as

$$\Phi_\nu(v) = \int_{\mathbb{R}^p} \mathrm{e}^{\mathrm{i}\langle v,w\rangle} \mathrm{d}\nu(w) \ .$$

Then, we can conclude that for Lebesgue-almost every $z \in \mathbb{R}^d$,

$$\lim_{k\to\infty} \Phi_{\mu_{\phi(k)}}(z) = \Phi_\mu(z) \ . \tag{S3}$$

We can now show that $(\mu_{\phi(k)})_{k\in\mathbb{N}} \xrightarrow{w} \mu$, *i.e.* by [3, Problem 1.11, Chapter 1] for any $f : \mathbb{R}^d \to \mathbb{R}$ continuous with compact support,

$$\lim_{n\to\infty} \int_{\mathbb{R}^d} f(z)\mathrm{d}\mu_n(z) = \int_{\mathbb{R}^d} f(z)\mathrm{d}\mu(z) \ . \tag{S4}$$

Let $f : \mathbb{R}^d \to \mathbb{R}$ be a continuous function with compact support and $\sigma > 0$. Consider the function $f_\sigma$ defined for any $x \in \mathbb{R}^d$ as

$$f_\sigma(x) = (2\pi\sigma^2)^{-d/2} \int_{\mathbb{R}^d} f(x-z) \exp\left(-\|z\|^2/2\sigma^2\right) \mathrm{dLeb}(z) = f * g_\sigma(x) \ ,$$

where $g_\sigma$ is the density of the $d$-dimensional Gaussian with covariance matrix $\sigma^2 \mathbf{I}_d$ and $*$ denotes the convolution product.

We first show that (S4) holds with $f_\sigma$ in place of $f$. Since for any $z \in \mathbb{R}^d$, $\mathbb{E}\left[\mathrm{e}^{\mathrm{i}\langle G,z\rangle}\right] = \mathrm{e}^{\mathrm{i}\langle \mathrm{m},z\rangle + (1/(2\sigma^2))\|z\|^2}$ if $G$ is a $d$-dimensional Gaussian random variable with zero mean and covariance matrix $(1/\sigma^2)\,\mathrm{I}_d$, by Fubini's theorem we get for any $k \in \mathbb{N}$

$$\begin{aligned}
\int_{\mathbb{R}^d} f_\sigma(z)\mathrm{d}\mu_{\phi(k)}(z) &= \int_{\mathbb{R}^d} \int_{\mathbb{R}^d} f(w)g_\sigma(z-w)\mathrm{d}w\mathrm{d}\mu_{\phi(k)}(z) \\
&= \int_{\mathbb{R}^d} \int_{\mathbb{R}^d} f(w)(2\pi\sigma^2)^{-d/2} \int_{\mathbb{R}^d} \mathrm{e}^{\mathrm{i}\langle z-w,x\rangle} g_{1/\sigma}(x)\mathrm{d}x\mathrm{d}w\mathrm{d}\mu_{\phi(k)}(z) \\
&= \int_{\mathbb{R}^d} \int_{\mathbb{R}^d} (2\pi\sigma^2)^{-d/2} f(w)\mathrm{e}^{-\mathrm{i}\langle w,x\rangle} g_{1/\sigma}(x)\Phi_{\mu_{\phi(k)}}(x)\mathrm{d}x\mathrm{d}w \\
&= (2\pi\sigma^2)^{-d/2} \int_{\mathbb{R}^d} \mathcal{F}[f](x)g_{1/\sigma}(x)\Phi_{\mu_{\phi(k)}}(x)\mathrm{d}x \ , \tag{S5}
\end{aligned}$$

where $\mathcal{F}[f](x) = \int_{\mathbb{R}^d} f(w)\mathrm{e}^{\mathrm{i}\langle w,x\rangle}\mathrm{d}w$ denotes the Fourier transform of $f$[1]. In an analogous manner, we prove that

$$\int_{\mathbb{R}^d} f_\sigma(z)\mathrm{d}\mu(z) = (2\pi\sigma^2)^{-d/2} \int_{\mathbb{R}^d} \mathcal{F}[f](x)g_{1/\sigma}(x)\Phi_\mu(x)\mathrm{d}x \ . \tag{S6}$$

Now, using that $\mathcal{F}[f]$ is bounded by $\int_{\mathbb{R}^d} |f(w)|\mathrm{d}w < +\infty$ since $f$ has compact support, we obtain that, for any $k \in \mathbb{N}$ and $x \in \mathbb{R}^d$,

$$\left|\mathcal{F}[f](x)g_{1/\sigma}(x)\Phi_{\mu_{\phi(k)}}(x)\right| \leq g_{1/\sigma}(x) \int_{\mathbb{R}^d} |f(w)|\mathrm{d}w$$

By (S3), (S5), (S6) and Lebesgue's Dominated Convergence Theorem, we obtain

$$\lim_{k\to\infty}\int_{\mathbb{R}^d}(2\pi\sigma^2)^{-d/2}\mathcal{F}[f](x)g_{1/\sigma}(x)\Phi_{\mu_{\phi(k)}}(x)\mathrm{d}x=\int_{\mathbb{R}^d}(2\pi\sigma^2)^{-d/2}\mathcal{F}[f](x)g_{1/\sigma}(x)\Phi_{\mu}(x)\mathrm{d}x$$

$$\lim_{k\to\infty}\int_{\mathbb{R}^d}f_\sigma(z)\mathrm{d}\mu_{\phi(k)}(z)=\int_{\mathbb{R}^d}f_\sigma(z)\mathrm{d}\mu(z)\ . \tag{S7}$$

We can now complete the proof of (S4). For any $\sigma>0$, we have

$$\left|\int_{\mathbb{R}^d}f(z)\mathrm{d}\mu_{\phi(k)}(z)-\int_{\mathbb{R}^d}f(z)\mathrm{d}\mu(z)\right|\leq 2\sup_{z\in\mathbb{R}^d}|f(z)-f_\sigma(z)|$$

$$+\left|\int_{\mathbb{R}^d}f_\sigma(z)\mathrm{d}\mu_{\phi(k)}(z)-\int_{\mathbb{R}^d}f_\sigma(z)\mathrm{d}\mu(z)\right|\ .$$

Therefore by (S7), for any $\sigma>0$, we get

$$\limsup_{n\to+\infty}\left|\int_{\mathbb{R}^d}f(z)\mathrm{d}\mu_{\phi(k)}(z)-\int_{\mathbb{R}^d}f(z)\mathrm{d}\mu(z)\right|\leq 2\sup_{z\in\mathbb{R}^d}|f(z)-f_\sigma(z)|\ .$$

Finally [9, Theorem 8.14-b] implies that $\lim_{\sigma\to 0}\sup_{z\in\mathbb{R}^d}|f_\sigma(z)-f(z)|=0$ which concludes the proof. $\qquad\square$

*Proof of Theorem 1.* Now, assume that

$$\lim_{k\to\infty}\mathbf{SW}_p(\mu_k,\mu)=0 \tag{S8}$$

and that $(\mu_k)_{k\in\mathbb{N}}$ does not converge weakly to $\mu$. Therefore, $\lim_{k\to\infty}\mathbf{d}_\mathcal{P}(\mu_k,\mu)\neq 0$, where $\mathbf{d}_\mathcal{P}$ denotes the Lévy-Prokhorov metric, and there exists $\epsilon>0$ and a subsequence $(\mu_{\psi(k)})_{k\in\mathbb{N}}$ with $\psi:\mathbb{N}\to\mathbb{N}$ increasing, such that for any $k\in\mathbb{N}$,

$$\mathbf{d}_\mathcal{P}(\mu_{\psi(k)},\mu)>\epsilon \tag{S9}$$

In addition, by Hölder's inequality, we know that $\mathbf{W}_1(\mu_k,\mu)\leq\mathbf{W}_p(\mu_k,\mu)$, thus $\mathbf{SW}_1(\mu_k,\mu)\leq\mathbf{SW}_p(\mu_k,\mu)$, and by (S8), $\lim_{k\to\infty}\mathbf{SW}_1(\mu_{\psi(k)},\mu)=0$. Then, according to Lemma S13, there exists a subsequence $(\mu_{\phi(\psi(k))})_{k\in\mathbb{N}}$ with $\phi:\mathbb{N}\to\mathbb{N}$ increasing, such that

$$\mu_{\phi(\psi(k))}\xrightarrow{w}\mu$$

which is equivalent to $\lim_{k\to\infty}\mathbf{d}_\mathcal{P}(\mu_{\phi(\psi(k))},\mu)=0$, thus contradicts (S9). We conclude that (S8) implies $(\mu_k)_{k\in\mathbb{N}}\xrightarrow{w}\mu$. $\qquad\square$

## 3.2 Minimum Sliced-Wasserstein estimators: Proof of Theorem 2

*Proof of Theorem 2.* This result is proved analogously to the proof of Theorem 2.1 in [6]. The key step is to show that the function $\theta\mapsto\mathbf{SW}_p(\hat{\mu}_n,\mu_\theta)$ epi-converges to $\theta\mapsto\mathbf{SW}_p(\mu_\star,\mu_\theta)$ $\mathbb{P}$-almost surely, and then apply Theorem 7.31 of [1] (recalled in Theorem S5).

First, by **A1** and Corollary 7, the map $\theta\mapsto\mathbf{SW}_p(\mu,\mu_\theta)$ is l.s.c. on $\Theta$ for any $\mu\in\mathcal{P}_p(\mathsf{Y})$. Therefore by **A3**, there exists $\theta_\star\in\Theta$ such that $\mathbf{SW}_p(\mu_\star,\mu_{\theta_\star})=\epsilon_\star$ and the set $\Theta_\epsilon^\star$ is non-empty as it contains $\theta_\star$, closed by lower semi-continuity of $\theta\mapsto\mathbf{SW}_p(\mu_\star,\mu_\theta)$, and bounded. $\Theta_\epsilon^\star$ is thus compact, and we conclude again by lower semi-continuity that the set $\mathrm{argmin}_{\theta\in\Theta}\mathbf{SW}_p(\mu_\star,\mu_\theta)$ is non-empty [10, Theorem 2.43].

Consider the event given by **A2**, $\mathsf{E}\in\mathcal{F}$ such that $\mathbb{P}(\mathsf{E})=1$ and for any $\omega\in\mathsf{E}$, $\lim_{n\to\infty}\mathbf{SW}_p(\hat{\mu}_n(\omega),\mu_\star)=0$. Then, we prove that $\theta\mapsto\mathbf{SW}_p(\hat{\mu}_n,\mu_\theta)$ epi-converges to $\theta\mapsto\mathbf{SW}_p(\mu_\star,\mu_\theta)$ $\mathbb{P}$-almost surely using the characterization in [1, Proposition 7.29], *i.e.* we verify that, for any $\omega\in\mathsf{E}$, the two conditions below hold: for every compact set $\mathsf{K}\subset\Theta$ and every open set $\mathsf{O}\subset\Theta$,

$$\liminf_{n\to\infty}\inf_{\theta\in\mathsf{K}}\mathbf{SW}_p(\hat{\mu}_n(\omega),\mu_\theta)\geq\inf_{\theta\in\mathsf{K}}\mathbf{SW}_p(\mu_\star,\mu_\theta)$$

$$\limsup_{n\to\infty}\inf_{\theta\in\mathsf{O}}\mathbf{SW}_p(\hat{\mu}_n(\omega),\mu_\theta)\leq\inf_{\theta\in\mathsf{O}}\mathbf{SW}_p(\mu_\star,\mu_\theta)\ . \tag{S10}$$

We fix $\omega$ in E. Let $\mathsf{K} \subset \Theta$ be a compact set. By lower semi-continuity of $\theta \mapsto \mathbf{SW}_p(\hat{\mu}_n(\omega), \mu_\theta)$, there exists $\theta_n = \theta_n(\omega) \in \mathsf{K}$ such that for any $n \in \mathbb{N}$, $\inf_{\theta \in \mathsf{K}} \mathbf{SW}_p(\hat{\mu}_n(\omega), \mu_\theta) = \mathbf{SW}_p(\hat{\mu}_n(\omega), \mu_{\theta_n})$.

We consider the subsequence $(\hat{\mu}_{\phi(n)})_{n \in \mathbb{N}}$ where $\phi : \mathbb{N} \to \mathbb{N}$ is increasing such that $\mathbf{SW}_p(\hat{\mu}_{\phi(n)}(\omega), \mu_{\theta_{\phi(n)}})$ converges to $\liminf_{n \to \infty} \mathbf{SW}_p(\hat{\mu}_n(\omega), \mu_{\theta_n}) = \liminf_{n \to \infty} \inf_{\theta \in \mathsf{K}} \mathbf{SW}_p(\hat{\mu}_n(\omega), \mu_\theta)$. Since $\mathsf{K}$ is compact, there also exists an increasing function $\psi : \mathbb{N} \to \mathbb{N}$ such that, for $\bar{\theta} \in \mathsf{K}$, $\lim_{n \to \infty} \rho_\Theta(\theta_{\psi(\phi(n))}, \bar{\theta}) = 0$. Therefore, we have

$$
\begin{aligned}
\liminf_{n \to \infty} \inf_{\theta \in \mathsf{K}} \mathbf{SW}_p(\hat{\mu}_n(\omega), \mu_\theta) &= \lim_{n \to \infty} \mathbf{SW}_p(\hat{\mu}_{\phi(n)}(\omega), \mu_{\theta_{\phi(n)}}) \\
&= \lim_{n \to \infty} \mathbf{SW}_p(\hat{\mu}_{\psi(\phi(n))}(\omega), \mu_{\theta_{\psi(\phi(n))}}) \\
&= \liminf_{n \to \infty} \mathbf{SW}_p(\hat{\mu}_{\psi(\phi(n))}(\omega), \mu_{\theta_{\psi(\phi(n))}}) \\
&\geq \mathbf{SW}_p(\mu_\star, \mu_{\bar{\theta}}) \\
&\geq \inf_{\theta \in \mathsf{K}} \mathbf{SW}_p(\mu_\star, \mu_\theta) \, ,
\end{aligned}
\tag{S11}
$$

where (S11) is obtained by lower semi-continuity since $\hat{\mu}_{\psi(\phi(n))}(\omega) \xrightarrow{w} \mu_\star$ by **A2** and Theorem 1, and $\mu_{\theta_{\psi(\phi(n))}} \xrightarrow{w} \mu_{\bar{\theta}}$ by **A1**. We conclude that the first condition in (S10) holds.

Now, we fix $\mathsf{O} \subset \Theta$ open. By definition of the infimum, there exists a sequence $(\theta_n)_{n \in \mathbb{N}}$ in $\mathsf{O}$ such that $\{\mathbf{SW}_p(\mu_\star, \mu_{\theta_n})\}_{n \in \mathbb{N}}$ converges to $\inf_{\theta \in \mathsf{O}} \mathbf{SW}_p(\mu_\star, \mu_\theta)$. For any $n \in \mathbb{N}$, $\inf_{\theta \in \mathsf{O}} \mathbf{SW}_p(\hat{\mu}_n(\omega), \mu_\theta) \leq \mathbf{SW}_p(\hat{\mu}_n(\omega), \mu_{\theta_n})$. Therefore,

$$
\begin{aligned}
\limsup_{n \to \infty} \inf_{\theta \in \mathsf{O}} \mathbf{SW}_p(\hat{\mu}_n(\omega), \mu_\theta) &\leq \limsup_{n \to \infty} \mathbf{SW}_p(\hat{\mu}_n(\omega), \mu_{\theta_n}) \\
&\leq \limsup_{n \to \infty} \left( \mathbf{SW}_p(\hat{\mu}_n(\omega), \mu_\star) + \mathbf{SW}_p(\mu_\star, \mu_{\theta_n}) \right) \text{ by the triangle inequality} \\
&\leq \limsup_{n \to \infty} \mathbf{SW}_p(\mu_\star, \mu_{\theta_n}) \text{ by } \mathbf{A2} \\
&= \inf_{\theta \in \mathsf{O}} \mathbf{SW}_p(\mu_\star, \mu_\theta) \text{ by definition of } (\theta_n)_{n \in \mathbb{N}} \, .
\end{aligned}
$$

This shows that the second condition in (S10) holds, and hence, the sequence of functions $\theta \mapsto \mathbf{SW}_p(\hat{\mu}_n(\omega), \mu_\theta)$ epi-converges to $\theta \mapsto \mathbf{SW}_p(\mu_\star, \mu_\theta)$.

Now, we apply Theorem 7.31 of [1]. First, by [1, Theorem 7.31(b)], (9) immediately follows from the epi-convergence of $\theta \mapsto \mathbf{SW}_p(\hat{\mu}_n(\omega), \mu_\theta)$ to $\theta \mapsto \mathbf{SW}_p(\mu_\star, \mu_\theta)$.

Next, we show that [1, Theorem 7.31(a)] can be applied showing that for any $\eta > 0$ there exists a compact set $\mathsf{B} \subset \Theta$ and $N \in \mathbb{N}$ such that, for all $n \geq N$,

$$
\inf_{\theta \in \mathsf{B}} \mathbf{SW}_p(\hat{\mu}_n(\omega), \mu_\theta) \leq \inf_{\theta \in \Theta} \mathbf{SW}_p(\hat{\mu}_n(\omega), \mu_\theta) + \eta \, .
\tag{S12}
$$

In fact, we simply show that there exists a compact set $\mathsf{B} \subset \Theta$ and $N \in \mathbb{N}$ such that, for all $n \geq N$, $\inf_{\theta \in \mathsf{B}} \mathbf{SW}_p(\hat{\mu}_n(\omega), \mu_\theta) = \inf_{\theta \in \Theta} \mathbf{SW}_p(\hat{\mu}_n(\omega), \mu_\theta)$.

On one hand, the second condition in (S10) gives us

$$
\limsup_{n \to \infty} \inf_{\theta \in \Theta} \mathbf{SW}_p(\hat{\mu}_n(\omega), \mu_\theta) \leq \inf_{\theta \in \Theta} \mathbf{SW}_p(\mu_\star, \mu_\theta) = \epsilon_\star \, .
$$

We deduce that there exists $n_{\epsilon/4}(\omega)$ such that, for $n \geq n_{\epsilon/4}(\omega)$, $\inf_{\theta \in \Theta} \mathbf{SW}_p(\hat{\mu}_n(\omega), \mu_\theta) \leq \epsilon_\star + \epsilon/4$, where $\epsilon$ is given by **A3**. As $n \geq n_{\epsilon/4}(\omega)$, the set $\widehat{\Theta}_{\epsilon/2} = \{\theta \in \Theta : \mathbf{SW}_p(\hat{\mu}_n(\omega), \mu_\theta) \leq \epsilon_\star + \frac{\epsilon}{2}\}$ is non-empty as it contains $\theta^*$ defined as $\mathbf{SW}_p(\hat{\mu}_n(\omega), \mu_{\theta^*}) = \inf_{\theta \in \Theta} \mathbf{SW}_p(\hat{\mu}_n(\omega), \mu_\theta)$.

On the other hand, by **A2**, there exists $n_{\epsilon/2}(\omega)$ such that, for $n \geq n_{\epsilon/2}(\omega)$,

$$
\mathbf{SW}_p(\hat{\mu}_n(\omega), \mu_\star) \leq \frac{\epsilon}{2} \, .
\tag{S13}
$$

Let $n \geq n_*(\omega) = \max\{n_{\epsilon/4}(\omega), n_{\epsilon/2}(\omega)\}$ and $\theta \in \widehat{\Theta}_{\epsilon/2}$. By the triangle inequality,

$$
\begin{aligned}
\mathbf{SW}_p(\mu_\star, \mu_\theta) &\leq \mathbf{SW}_p(\hat{\mu}_n(\omega), \mu_\star) + \mathbf{SW}_p(\hat{\mu}_n(\omega), \mu_\theta) \\
&\leq \epsilon_\star + \epsilon \quad \text{since } \theta \in \widehat{\Theta}_{\epsilon/2} \text{ and by (S13)}
\end{aligned}
$$

This means that, when $n \geq n_*(\omega)$, $\widehat{\Theta}_{\epsilon/2} \subset \Theta_\epsilon^\star$, and since $\inf_{\theta \in \Theta} \mathbf{SW}_p(\hat{\mu}_n(\omega), \mu_\theta)$ is attained in $\widehat{\Theta}_{\epsilon/2}$, we have

$$\inf_{\theta \in \Theta_\epsilon^\star} \mathbf{SW}_p(\hat{\mu}_n(\omega), \mu_\theta) = \inf_{\theta \in \Theta} \mathbf{SW}_p(\hat{\mu}_n(\omega), \mu_\theta) \,. \tag{S14}$$

As shown in the first part of the proof $\Theta_\epsilon^\star$ is compact and then by [1, Theorem 7.31(a)], (8) is a direct consequence of (S12)-(S14) and the epi-convergence of $\theta \mapsto \mathbf{SW}_p(\hat{\mu}_n(\omega), \mu_\theta)$ to $\theta \mapsto \mathbf{SW}_p(\mu_\star, \mu_\theta)$.

Finally, by the same reasoning that was done earlier in this proof for $\operatorname{argmin}_{\theta \in \Theta} \mathbf{SW}_p(\mu_\star, \mu_\theta)$, the set $\operatorname{argmin}_{\theta \in \Theta} \mathbf{SW}_p(\hat{\mu}_n(\omega), \mu_\theta)$ is non-empty for $n \geq n_*(\omega)$.

$\square$

### 3.3 Existence and consistency of the MESWE: Proof of Theorem 3

*Proof of Theorem 3.* This result is proved analogously to the proof of [6, Theorem 2.4]. The key step is to show that the function $\theta \mapsto \mathbb{E}[\mathbf{SW}_p(\hat{\mu}_n, \hat{\mu}_{\theta, m(n)})|Y_{1:n}]$ epi-converges to $\theta \mapsto \mathbb{E}[\mathbf{SW}_p(\mu_\star, \mu_\theta)|Y_{1:n}]$, and then apply [1, Theorem 7.31], which we recall in Theorem S5.

First, since we assume **A**1 and **A**3, we can apply the same reasoning as in the proof of Theorem 2 to show that the set $\operatorname{argmin}_{\theta \in \Theta} \mathbf{SW}_p(\mu_\star, \mu_\theta)$ is non-empty.

Consider the event given by **A**2, $\mathsf{E} \in \mathcal{F}$ such that $\mathbb{P}(\mathsf{E}) = 1$ and for any $\omega \in \mathsf{E}$, $\lim_{n \to \infty} \mathbf{SW}_p(\hat{\mu}_n(\omega), \mu_\star) = 0$. Then, we prove that $\theta \mapsto \mathbb{E}[\mathbf{SW}_p(\hat{\mu}_n, \hat{\mu}_{\theta, m(n)})|Y_{1:n}]$ epi-converges to $\theta \mapsto \mathbf{SW}_p(\mu_\star, \mu_\theta)$ $\mathbb{P}$-almost surely using the characterization of [1, Proposition 7.29], *i.e.* we verify that, for any $\omega \in \mathsf{E}$, the two conditions below hold: for every compact set $\mathsf{K} \subset \Theta$ and for every open set $\mathsf{O} \subset \Theta$,

$$\liminf_{n \to +\infty} \inf_{\theta \in \mathsf{K}} \mathbb{E}\left[\mathbf{SW}_p(\hat{\mu}_n(\omega), \hat{\mu}_{\theta, m(n)})\big|Y_{1:n}\right] \geq \inf_{\theta \in \mathsf{K}} \mathbf{SW}_p(\mu_\star, \mu_\theta)$$

$$\limsup_{n \to +\infty} \inf_{\theta \in \mathsf{O}} \mathbb{E}\left[\mathbf{SW}_p(\hat{\mu}_n(\omega), \hat{\mu}_{\theta, m(n)})\big|Y_{1:n}\right] \leq \inf_{\theta \in \mathsf{O}} \mathbf{SW}_p(\mu_\star, \mu_\theta) \tag{S15}$$

We fix $\omega$ in $\mathsf{E}$. Let $\mathsf{K} \subset \Theta$ be a compact set. By **A**1 and Corollary 9, the mapping $\theta \mapsto \mathbb{E}[\mathbf{SW}_p(\hat{\mu}_n(\omega), \hat{\mu}_{\theta, m(n)})|Y_{1:n}]$ is l.s.c., so there exists $\theta_n = \theta_n(\omega) \in \mathsf{K}$ such that for any $n \in \mathbb{N}$, $\inf_{\theta \in \mathsf{K}} \mathbb{E}\left[\mathbf{SW}_p(\hat{\mu}_n(\omega), \hat{\mu}_{\theta, m(n)})\big|Y_{1:n}\right] = \mathbb{E}\left[\mathbf{SW}_p(\hat{\mu}_n(\omega), \hat{\mu}_{\theta_n, m(n)})\big|Y_{1:n}\right]$.

We consider the subsequence $(\hat{\mu}_{\phi(n)})_{n \in \mathbb{N}}$ where $\phi : \mathbb{N} \to \mathbb{N}$ is increasing such that $\mathbb{E}[\mathbf{SW}_p(\hat{\mu}_{\phi(n)}(\omega), \hat{\mu}_{\theta_{\phi(n)}, m(\phi(n))})|Y_{1:n}]$ converges to $\liminf_{n \to \infty} \mathbb{E}[\mathbf{SW}_p(\hat{\mu}_n(\omega), \hat{\mu}_{\theta_{n,m(n)}})|Y_{1:n}] = \liminf_{n \to \infty} \inf_{\theta \in \mathsf{K}} \mathbb{E}[\mathbf{SW}_p(\hat{\mu}_n(\omega), \hat{\mu}_{\theta, m(n)})|Y_{1:n}]$. Since $\mathsf{K}$ is compact, there also exists an increasing function $\psi : \mathbb{N} \to \mathbb{N}$ such that, for $\bar{\theta} \in \mathsf{K}$, $\lim_{n \to \infty} \rho_\Theta(\theta_{\psi(\phi(n))}, \bar{\theta}) = 0$. Therefore, we have:

$$\liminf_{n \to \infty} \inf_{\theta \in \mathsf{K}} \mathbb{E}\left[\mathbf{SW}_p(\hat{\mu}_n(\omega), \hat{\mu}_{\theta, m(n)})\big|Y_{1:n}\right]$$

$$= \lim_{n \to \infty} \mathbb{E}\left[\mathbf{SW}_p(\hat{\mu}_{\phi(n)}(\omega), \hat{\mu}_{\theta_{\phi(n)}, m(\phi(n))})\big|Y_{1:n}\right]$$

$$= \lim_{n \to \infty} \mathbb{E}\left[\mathbf{SW}_p(\hat{\mu}_{\psi(\phi(n))}(\omega), \hat{\mu}_{\theta_{\psi(\phi(n))}, m(\psi(\phi(n)))})\big|Y_{1:n}\right]$$

$$= \liminf_{n \to \infty} \mathbb{E}\left[\mathbf{SW}_p(\hat{\mu}_{\psi(\phi(n))}(\omega), \hat{\mu}_{\theta_{\psi(\phi(n))}, m(\psi(\phi(n)))})\big|Y_{1:n}\right]$$

$$\geq \liminf_{n \to \infty} \left\{ \mathbf{SW}_p(\hat{\mu}_{\psi(\phi(n))}(\omega), \mu_{\theta_{\psi(\phi(n))}}) - \mathbb{E}\left[\mathbf{SW}_p(\mu_{\theta_{\psi(\phi(n))}}, \hat{\mu}_{\theta_{\psi(\phi(n))}, m(\psi(\phi(n)))})\big|Y_{1:n}\right] \right\}$$
$$\tag{S16}$$

$$\geq \liminf_{n \to \infty} \mathbf{SW}_p(\hat{\mu}_{\psi(\phi(n))}(\omega), \mu_{\theta_{\psi(\phi(n))}}) - \limsup_{n \to \infty} \mathbb{E}\left[\mathbf{SW}_p(\mu_{\theta_{\psi(\phi(n))}}, \hat{\mu}_{\theta_{\psi(\phi(n))}, m(\psi(\phi(n)))})\big|Y_{1:n}\right]$$

$$\geq \mathbf{SW}_p(\mu_\star, \mu_{\bar{\theta}}) \tag{S17}$$

$$\geq \inf_{\theta \in \mathsf{K}} \mathbf{SW}_p(\mu_\star, \mu_\theta)$$

where (S16) follows from the triangle inequality, and (S17) is obtained on one hand by lower semi-continuity since $\hat{\mu}_{\psi(\phi(n))}(\omega) \xrightarrow{w} \mu_\star$ by **A**2 and Theorem 1 and $\mu_{\theta_{\psi(\phi(n))}} \xrightarrow{w} \mu_{\bar{\theta}}$ by **A**1, and on the

other hand by **A4** which gives $\limsup_{n\to\infty} \mathbb{E}[\mathbf{SW}_p(\mu_{\theta_{\psi(\phi(n))}}, \hat{\mu}_{\theta_{\psi(\phi(n))}, m(\psi(\phi(n)))}) | Y_{1:n}] = 0$. We conclude that the first condition in (S15) holds.

Now, we fix $\mathsf{O} \subset \Theta$ open. By definition of the infimum, there exists a sequence $(\theta_n)_{n\in\mathbb{N}}$ in $\mathsf{O}$ such that $\mathbf{SW}_p(\mu_\star, \mu_{\theta_n})$ converges to $\inf_{\theta\in\mathsf{O}} \mathbf{SW}_p(\mu_\star, \mu_\theta)$. For any $n \in \mathbb{N}$, $\inf_{\theta\in\mathsf{O}} \mathbb{E}\left[\mathbf{SW}_p(\hat{\mu}_n(\omega), \hat{\mu}_{\theta, m(n)}) | Y_{1:n}\right] \leq \mathbb{E}\left[\mathbf{SW}_p(\hat{\mu}_n(\omega), \hat{\mu}_{\theta_n, m(n)}) | Y_{1:n}\right]$. Therefore,

$$\limsup_{n\to\infty} \inf_{\theta\in\mathsf{O}} \mathbb{E}\left[\mathbf{SW}_p(\hat{\mu}_n(\omega), \hat{\mu}_{\theta, m(n)}) \big| Y_{1:n}\right] \leq \limsup_{n\to\infty} \mathbb{E}\left[\mathbf{SW}_p(\hat{\mu}_n(\omega), \hat{\mu}_{\theta_n, m(n)}) \big| Y_{1:n}\right]$$

$$\leq \limsup_{n\to\infty} \left\{\mathbf{SW}_p(\hat{\mu}_n(\omega), \mu_\star) + \mathbf{SW}_p(\mu_\star, \mu_{\theta_n}) + \mathbb{E}\left[\mathbf{SW}_p(\mu_{\theta_n}, \hat{\mu}_{\theta_n, m(n)}) \big| Y_{1:n}\right]\right\}$$

$$\text{by the triangle inequality}$$

$$= \limsup_{n\to\infty} \mathbf{SW}_p(\mu_\star, \mu_{\theta_n}) \quad \text{by } \mathbf{A2} \text{ and } \mathbf{A4}$$

$$= \inf_{\theta\in\mathsf{O}} \mathbf{SW}_p(\mu_\star, \mu_\theta) \quad \text{by definition of } (\theta_n)_{n\in\mathbb{N}}.$$

This shows that the second condition in (S15) holds, and hence, the sequence of functions $\theta \mapsto \mathbb{E}\left[\mathbf{SW}_p(\hat{\mu}_n(\omega), \hat{\mu}_{\theta, m(n)}) \big| Y_{1:n}\right]$ epi-converges to $\theta \mapsto \mathbf{SW}_p(\mu_\star, \mu_\theta)$.

Now, we apply Theorem 7.31 of [1]. First, by [1, Theorem 7.31(b)], (11) immediately follows from the epi-convergence of $\theta \mapsto \mathbb{E}\left[\mathbf{SW}_p(\hat{\mu}_n(\omega), \hat{\mu}_{\theta, m(n)}) \big| Y_{1:n}\right]$ to $\theta \mapsto \mathbf{SW}_p(\mu_\star, \mu_\theta)$.

Next, we show that [1, Theorem 7.31(a)] holds by finding, for any $\eta > 0$, a compact set $\mathsf{B} \subset \Theta$ and $N \in \mathbb{N}$ such that, for all $n \geq N$,

$$\inf_{\theta\in\mathsf{B}} \mathbb{E}\left[\mathbf{SW}_p(\hat{\mu}_n(\omega), \hat{\mu}_{\theta, m(n)}) \big| Y_{1:n}\right] \leq \inf_{\theta\in\Theta} \mathbb{E}\left[\mathbf{SW}_p(\hat{\mu}_n(\omega), \hat{\mu}_{\theta, m(n)}) \big| Y_{1:n}\right] + \eta .$$

In fact, we simply show that there exists a compact set $\mathsf{B} \subset \Theta$ and $N \in \mathbb{N}$ such that, for all $n \geq N$, $\inf_{\theta\in\mathsf{B}} \mathbb{E}\left[\mathbf{SW}_p(\hat{\mu}_n(\omega), \hat{\mu}_{\theta, m(n)}) \big| Y_{1:n}\right] = \inf_{\theta\in\Theta} \mathbb{E}\left[\mathbf{SW}_p(\hat{\mu}_n(\omega), \hat{\mu}_{\theta, m(n)}) \big| Y_{1:n}\right]$.

On one hand, the second condition in (S15) gives us

$$\limsup_{n\to\infty} \inf_{\theta\in\Theta} \mathbb{E}\left[\mathbf{SW}_p(\hat{\mu}_n(\omega), \hat{\mu}_{\theta, m(n)}) \big| Y_{1:n}\right] \leq \inf_{\theta\in\Theta} \mathbf{SW}_p(\mu_\star, \mu_\theta) = \epsilon_\star .$$

We deduce that there exists $n_{\epsilon/6}(\omega)$ such that, for $n \geq n_{\epsilon/6}(\omega)$,

$$\inf_{\theta\in\Theta} \mathbb{E}\left[\mathbf{SW}_p(\hat{\mu}_n(\omega), \hat{\mu}_{\theta, m(n)}) \big| Y_{1:n}\right] \leq \epsilon_\star + \frac{\epsilon}{6},$$

with the $\epsilon$ of **A3**. When $n \geq n_{\epsilon/6}(\omega)$, the set $\widehat{\Theta}_{\epsilon/3} = \{\theta \in \Theta : \mathbb{E}[\mathbf{SW}_p(\hat{\mu}_n(\omega), \hat{\mu}_{\theta, m(n)}) | Y_{1:n}] \leq \epsilon_\star + \frac{\epsilon}{3}\}$ is non-empty as it contains $\theta^*$ defined as $\mathbb{E}\left[\mathbf{SW}_p(\hat{\mu}_n(\omega), \hat{\mu}_{\theta^*, m(n)}) \big| Y_{1:n}\right] = \inf_{\theta\in\Theta} \mathbb{E}\left[\mathbf{SW}_p(\hat{\mu}_n(\omega), \hat{\mu}_{\theta, m(n)}) \big| Y_{1:n}\right]$.

On the other hand, by **A2**, there exists $n_{\epsilon/3}(\omega)$ such that, for $n \geq n_{\epsilon/3}(\omega)$,

$$\mathbf{SW}_p(\hat{\mu}_n(\omega), \mu_\star) \leq \frac{\epsilon}{3} . \tag{S18}$$

Finally, by **A4**, there exists $n'_{\epsilon/3}(\omega)$ such that, for $n \geq n'_{\epsilon/3}(\omega)$,

$$\mathbb{E}\left[\mathbf{SW}_p(\mu_\theta, \hat{\mu}_{\theta, m(n)}) \big| Y_{1:n}\right] \leq \frac{\epsilon}{3} . \tag{S19}$$

Let $n \geq n_*(\omega) = \max\{n_{\epsilon/6}(\omega), n_{\epsilon/3}(\omega), n'_{\epsilon/3}(\omega)\}$ and $\theta \in \widehat{\Theta}_{\epsilon/3}$. By the triangle inequality,

$$\mathbf{SW}_p(\mu_\star, \mu_\theta) \leq \mathbf{SW}_p(\hat{\mu}_n(\omega), \mu_\star) + \mathbb{E}\left[\mathbf{SW}_p(\hat{\mu}_n(\omega), \hat{\mu}_{\theta, m(n)}) \big| Y_{1:n}\right] + \mathbb{E}\left[\mathbf{SW}_p(\mu_\theta, \hat{\mu}_{\theta, m(n)}) \big| Y_{1:n}\right]$$

$$\leq \epsilon_\star + \epsilon \quad \text{since } \theta \in \widehat{\Theta}_{\epsilon/3} \text{ and by (S18) and (S19)}$$

This means that, when $n \geq n_*(\omega)$, $\widehat{\Theta}_{\epsilon/3} \subset \Theta_\epsilon^\star$ with $\Theta_\epsilon^\star$ as defined in **A3**, and since $\inf_{\theta\in\Theta} \mathbb{E}\left[\mathbf{SW}_p(\hat{\mu}_n(\omega), \hat{\mu}_{\theta, m(n)}) \big| Y_{1:n}\right]$ is attained in $\widehat{\Theta}_{\epsilon/3}$, we have

$$\inf_{\theta\in\Theta_\epsilon^\star} \mathbb{E}\left[\mathbf{SW}_p(\hat{\mu}_n(\omega), \hat{\mu}_{\theta, m(n)}) \big| Y_{1:n}\right] = \inf_{\theta\in\Theta} \mathbb{E}\left[\mathbf{SW}_p(\hat{\mu}_n(\omega), \hat{\mu}_{\theta, m(n)}) \big| Y_{1:n}\right] . \tag{S20}$$

By [1, Theorem 7.31(a)], (10) is a direct consequence of (S20) and the epi-convergence of $\theta \mapsto \mathbb{E}\left[\mathbf{SW}_p(\hat{\mu}_n(\omega), \hat{\mu}_{\theta, m(n)}) \middle| Y_{1:n}\right]$ to $\theta \mapsto \mathbf{SW}_p(\mu_\star, \mu_\theta)$.

Finally, by the same reasoning that was done earlier in this proof for $\operatorname{argmin}_{\theta \in \Theta} \mathbf{SW}_p(\mu_\star, \mu_\theta)$, the set $\operatorname{argmin}_{\theta \in \Theta} \mathbb{E}\left[\mathbf{SW}_p(\hat{\mu}_n(\omega), \hat{\mu}_{\theta, m(n)}) \middle| Y_{1:n}\right]$ is non-empty for $n \geq n_*(\omega)$.

$\square$

### 3.4 Convergence of the MESWE to the MSWE: Proof of Theorem 4

*Proof of Theorem 4.* Here again, the result follows from applying [1, Theorem 7.31], paraphrased in Theorem S5.

First, by **A1** and Corollary 7, the map $\theta \mapsto \mathbf{SW}_p(\hat{\mu}_n, \mu_\theta)$ is l.s.c. on $\Theta$. Therefore, there exists $\theta_n \in \Theta$ such that $\mathbf{SW}_p(\hat{\mu}_n, \mu_{\theta_n}) = \epsilon_n$. The set $\Theta_{\epsilon, n}$ with the $\epsilon$ from **A5** is non-empty as it contains $\theta_n$, closed by lower semi-continuity of $\theta \mapsto \mathbf{SW}_p(\hat{\mu}_n, \mu_\theta)$, and bounded. $\Theta_{\epsilon, n}$ is thus compact, and we conclude again by lower semi-continuity that the set $\operatorname{argmin}_{\theta \in \Theta} \mathbf{SW}_p(\hat{\mu}_n, \mu_\theta)$ is non-empty [10, Theorem 2.43].

Then, we prove that $\theta \mapsto \mathbb{E}[\mathbf{SW}_p(\hat{\mu}_n, \hat{\mu}_{\theta, m})|Y_{1:n}]$ epi-converges to $\theta \mapsto \mathbf{SW}_p(\hat{\mu}_n, \mu_\theta)$ as $m \to \infty$ using the characterization in [1, Proposition 7.29], *i.e.* we verify that: for every compact set $\mathsf{K} \subset \Theta$ and every open set $\mathsf{O} \subset \Theta$,

$$\begin{aligned}
\liminf_{m \to \infty} \inf_{\theta \in \mathsf{K}} \mathbb{E}\left[\mathbf{SW}_p(\hat{\mu}_n, \hat{\mu}_{\theta, m})|Y_{1:n}\right] &\geq \inf_{\theta \in \mathsf{K}} \mathbf{SW}_p(\hat{\mu}_n, \mu_\theta) \\
\limsup_{m \to \infty} \inf_{\theta \in \mathsf{O}} \mathbb{E}\left[\mathbf{SW}_p(\hat{\mu}_n, \hat{\mu}_{\theta, m})|Y_{1:n}\right] &\leq \inf_{\theta \in \mathsf{O}} \mathbf{SW}_p(\hat{\mu}_n, \mu_\theta) .
\end{aligned} \tag{S21}$$

Let $\mathsf{K} \subset \Theta$ be a compact set. By **A1** and Corollary 9, for any $m \in \mathbb{N}$, the map $\theta \mapsto \mathbb{E}[\mathbf{SW}_p(\hat{\mu}_n, \hat{\mu}_{\theta, m})|Y_{1:n}]$ is l.s.c., so there exists $\theta_m \in \mathsf{K}$ such that $\inf_{\theta \in \mathsf{K}} \mathbb{E}[\mathbf{SW}_p(\hat{\mu}_n, \hat{\mu}_{\theta, m})|Y_{1:n}] = \mathbb{E}[\mathbf{SW}_p(\hat{\mu}_n, \hat{\mu}_{\theta_m, m})|Y_{1:n}]$.

We consider the subsequence $\{\hat{\mu}_{\theta_{\phi(m)}, \phi(m)}\}_{m \in \mathbb{N}}$ where $\phi : \mathbb{N} \to \mathbb{N}$ is increasing such that $\mathbb{E}[\mathbf{SW}_p(\hat{\mu}_n, \hat{\mu}_{\theta_{\phi(m)}, \phi(m)})|Y_{1:n}]$ converges to $\liminf_{m \to \infty} \mathbb{E}[\mathbf{SW}_p(\hat{\mu}_n, \hat{\mu}_{\theta_m, m})|Y_{1:n}] = \liminf_{m \to \infty} \inf_{\theta \in \mathsf{K}} \mathbb{E}[\mathbf{SW}_p(\hat{\mu}_n, \hat{\mu}_{\theta, m})|Y_{1:n}]$. Since $\mathsf{K}$ is compact, there also exists an increasing function $\psi : \mathbb{N} \to \mathbb{N}$ such that, for any $\bar{\theta} \in \mathsf{K}$, $\lim_{m \to \infty} \rho_\Theta(\theta_{\psi(\phi(m))}, \bar{\theta}) = 0$. Therefore, we have

$$\begin{aligned}
&\liminf_{m \to \infty} \inf_{\theta \in \mathsf{K}} \mathbb{E}\left[\mathbf{SW}_p(\hat{\mu}_n, \hat{\mu}_{\theta, m})|Y_{1:n}\right] \\
&= \lim_{m \to \infty} \mathbb{E}\left[\mathbf{SW}_p(\hat{\mu}_n, \hat{\mu}_{\theta_{\phi(m)}, \phi(m)}) \middle| Y_{1:n}\right] \\
&= \lim_{m \to \infty} \mathbb{E}\left[\mathbf{SW}_p(\hat{\mu}_n, \hat{\mu}_{\theta_{\psi(\phi(m))}, \psi(\phi(m))}) \middle| Y_{1:n}\right] \\
&= \liminf_{m \to \infty} \mathbb{E}\left[\mathbf{SW}_p(\hat{\mu}_n, \hat{\mu}_{\theta_{\psi(\phi(m))}, \psi(\phi(m))}) \middle| Y_{1:n}\right] \\
&\geq \liminf_{m \to \infty}[\mathbf{SW}_p(\hat{\mu}_n, \mu_{\theta_{\psi(\phi(m))}}) - \mathbb{E}\left[\mathbf{SW}_p(\mu_{\theta_{\psi(\phi(m))}}, \hat{\mu}_{\theta_{\psi(\phi(m))}, \psi(\phi(m))}) \middle| Y_{1:n}\right]] &\text{(S22)} \\
&\geq \liminf_{m \to \infty} \mathbf{SW}_p(\hat{\mu}_n, \mu_{\theta_{\psi(\phi(m))}}) - \limsup_{m \to \infty} \mathbb{E}\left[\mathbf{SW}_p(\mu_{\theta_{\psi(\phi(m))}}, \hat{\mu}_{\theta_{\psi(\phi(m))}, \psi(\phi(m))}) \middle| Y_{1:n}\right] \\
&\geq \mathbf{SW}_p(\hat{\mu}_n, \mu_{\bar{\theta}}) &\text{(S23)} \\
&\geq \inf_{\theta \in \mathsf{K}} \mathbf{SW}_p(\hat{\mu}_n, \mu_\theta)
\end{aligned}$$

where (S22) results from the triangle inequality and (S23) is obtained by **A4** on one hand and by lower semi-continuity on the other hand since $\mu_{\theta_{\psi(\phi(n))}} \xrightarrow{w} \mu_{\bar{\theta}}$ by **A1**. We conclude that the first condition in (S21) holds.

Now, we fix $\mathsf{O} \subset \Theta$ open. By definition of the infimum, there exists a sequence $(\theta_m)_{m \in \mathbb{N}}$ in $\mathsf{O}$ such that $\mathbf{SW}_p(\hat{\mu}_n, \hat{\mu}_{\theta_m, m})$ converges to $\inf_{\theta \in \mathsf{O}} \mathbf{SW}_p(\hat{\mu}_n, \hat{\mu}_{\theta, m})$. For any $m \in \mathbb{N}$,

$\inf_{\theta \in \mathsf{O}} \mathbb{E}\left[\mathbf{SW}_p(\hat{\mu}_n, \hat{\mu}_{\theta,m})|Y_{1:n}\right] \leq \mathbb{E}\left[\mathbf{SW}_p(\hat{\mu}_n, \mu_{\theta_m,m})|Y_{1:n}\right]$. Therefore,

$$\limsup_{m \to \infty} \inf_{\theta \in \mathsf{O}} \mathbb{E}\left[\mathbf{SW}_p(\hat{\mu}_n, \hat{\mu}_{\theta,m})|Y_{1:n}\right]$$

$$\leq \limsup_{m \to \infty} \mathbb{E}\left[\mathbf{SW}_p(\hat{\mu}_n, \hat{\mu}_{\theta_m,m})|Y_{1:n}\right]$$

$$\leq \limsup_{m \to \infty}[\mathbf{SW}_p(\hat{\mu}_n, \mu_{\theta_m}) + \mathbb{E}\left[\mathbf{SW}_p(\mu_{\theta_m}, \hat{\mu}_{\theta_m,m})|Y_{1:n}\right]] \quad \text{by the triangle inequality}$$

$$\leq \limsup_{m \to \infty} \mathbf{SW}_p(\hat{\mu}_n, \mu_{\theta_m}) \quad \text{by } \mathbf{A4}$$

$$= \inf_{\theta \in \mathsf{O}} \mathbf{SW}_p(\hat{\mu}_n, \mu_\theta) \quad \text{by definition of } (\theta_m)_{m \in \mathbb{N}}$$

This shows that the second condition in (S21) holds, and hence, the sequence of functions $\theta \mapsto \mathbb{E}\left[\mathbf{SW}_p(\hat{\mu}_n, \hat{\mu}_{\theta,m})|Y_{1:n}\right]$ epi-converges to $\theta \mapsto \mathbf{SW}_p(\hat{\mu}_n, \mu_\theta)$.

Now, we apply [1, Theorem 7.31]. By [1, Theorem 7.31(b)], (13) immediately follows from the epi-convergence of $\theta \mapsto \mathbb{E}\left[\mathbf{SW}_p(\hat{\mu}_n, \hat{\mu}_{\theta,m})|Y_{1:n}\right]$ to $\theta \mapsto \mathbf{SW}_p(\hat{\mu}_n, \mu_\theta)$.

Next, we show that [1, Theorem 7.31(a)] holds by finding for any $\eta > 0$ a compact set $\mathsf{B} \subset \Theta$ and $N \in \mathbb{N}$ such that, for all $n \geq N$,

$$\inf_{\theta \in \mathsf{B}} \mathbb{E}\left[\mathbf{SW}_p(\hat{\mu}_n, \hat{\mu}_{\theta,m})|Y_{1:n}\right] \leq \inf_{\theta \in \Theta} \mathbb{E}\left[\mathbf{SW}_p(\hat{\mu}_n, \hat{\mu}_{\theta,m})|Y_{1:n}\right] + \eta .$$

In fact, we simply show that there exists a compact set $\mathsf{B} \subset \Theta$ and $N \in \mathbb{N}$ such that, for all $n \geq N$, $\inf_{\theta \in \mathsf{B}} \mathbb{E}\left[\mathbf{SW}_p(\hat{\mu}_n, \hat{\mu}_{\theta,m})|Y_{1:n}\right] = \inf_{\theta \in \Theta} \mathbb{E}\left[\mathbf{SW}_p(\hat{\mu}_n, \hat{\mu}_{\theta,m})|Y_{1:n}\right]$. On one hand, the second condition in (S21) gives us

$$\limsup_{m \to \infty} \inf_{\theta \in \Theta} \mathbb{E}\left[\mathbf{SW}_p(\hat{\mu}_n, \hat{\mu}_{\theta,m})|Y_{1:n}\right] \leq \inf_{\theta \in \Theta} \mathbf{SW}_p(\hat{\mu}_n, \mu_\theta) = \epsilon_n .$$

We deduce that there exists $m_{\epsilon/4}$ such that, for $m \geq m_{\epsilon/4}$,

$$\inf_{\theta \in \Theta} \mathbb{E}\left[\mathbf{SW}_p(\hat{\mu}_n, \hat{\mu}_{\theta,m})|Y_{1:n}\right] \leq \epsilon_n + \frac{\epsilon}{4} . \tag{S24}$$

with the $\epsilon$ of $\mathbf{A5}$. When $m \geq m_{\epsilon/4}$, the set $\Theta_{\epsilon/2} = \{\theta \in \Theta : \mathbb{E}[\mathbf{SW}_p(\hat{\mu}_n, \hat{\mu}_{\theta,m})|Y_{1:n}] \leq \epsilon_n + \frac{\epsilon}{2}\}$ is non-empty as it contains $\theta^*$ defined as $\mathbb{E}[\mathbf{SW}_p(\hat{\mu}_n, \hat{\mu}_{\theta^*,m})|Y_{1:n}] = \inf_{\theta \in \Theta} \mathbb{E}[\mathbf{SW}_p(\hat{\mu}_n, \hat{\mu}_{\theta,m})|Y_{1:n}]$.

On the other hand, by $\mathbf{A4}$, there exists $m_{\epsilon/2}$ such that, for $m \geq m_{\epsilon/2}$,

$$\mathbb{E}\left[\mathbf{SW}_p(\mu_\theta, \hat{\mu}_{\theta,m})|Y_{1:n}\right] \leq \frac{\epsilon}{2} . \tag{S25}$$

Let $\theta$ belong to $\Theta_{\epsilon/2}$ and $m \geq m_* = \max\{m_{\epsilon/4}, m_{\epsilon/2}\}$. By the triangle inequality,

$$\mathbf{SW}_p(\hat{\mu}_n, \mu_\theta) \leq \mathbb{E}\left[\mathbf{SW}_p(\hat{\mu}_n, \hat{\mu}_{\theta,m})|Y_{1:n}\right] + \mathbb{E}\left[\mathbf{SW}_p(\mu_\theta, \hat{\mu}_{\theta,m})|Y_{1:n}\right]$$

$$\leq \epsilon_n + \epsilon \quad \text{since } \theta \in \Theta_{\epsilon/2} \text{ and by (S25)}$$

This means that, when $m \geq m_*$, $\Theta_{\epsilon/2} \subset \Theta_{\epsilon,n}$, and since $\inf_{\theta \in \Theta} \mathbb{E}\left[\mathbf{SW}_p(\hat{\mu}_n, \hat{\mu}_{\theta,m})|Y_{1:n}\right]$ is attained in $\Theta_{\epsilon/2}$,

$$\inf_{\theta \in \Theta_{\epsilon,n}} \mathbb{E}\left[\mathbf{SW}_p(\hat{\mu}_n, \hat{\mu}_{\theta,m})|Y_{1:n}\right] = \inf_{\theta \in \Theta} \mathbb{E}\left[\mathbf{SW}_p(\hat{\mu}_n, \hat{\mu}_{\theta,m})|Y_{1:n}\right] . \tag{S26}$$

By [1, Theorem 7.31(a)], (12) is a direct consequence of (S26) and the epiconvergence of $\theta \mapsto \mathbb{E}\left[\mathbf{SW}_p(\hat{\mu}_n(\omega), \hat{\mu}_{\theta,m})|Y_{1:n}\right]$ to $\theta \mapsto \mathbf{SW}_p(\hat{\mu}_n, \mu_\theta)$.

Finally, by the same reasoning that was done earlier in this proof for $\operatorname{argmin}_{\theta \in \Theta}\mathbf{SW}_p(\hat{\mu}_n, \mu_\theta)$, the set $\operatorname{argmin}_{\theta \in \Theta}\mathbb{E}\left[\mathbf{SW}_p(\hat{\mu}_n, \hat{\mu}_{\theta,m})|Y_{1:n}\right]$ is non-empty for $m \geq m_*$.

$\square$

### 3.5 Proof of Rate of convergence and asymptotic distribution: Proof of Theorem 5 and Theorem 6

*Proof of Theorem 5 and Theorem 6.* The proof of Theorem 5 and Theorem 6 consists in showing that the conditions of Theorem 4.2 and Theorem 7.2 in [11] respectively are satisfied: conditions (i), (ii) and (iii) follow from $\mathbf{A6}$, $\mathbf{A7}$ and $\mathbf{A8}$. $\square$

# 4 Computational Aspects

The MSWE and MESWE are in general computationnally intractable, partly because the Sliced-Wasserstein distance requires an integration over infinitely many projections. In this section, we review the numerical methods used to approximate these two estimators.

**Approximation of $\mathbf{SW}_p$:** We recall the definition of the SW distance below.

$$\mathbf{SW}_p^p(\mu, \nu) = \int_{\mathbb{S}^{d-1}} \mathbf{W}_p^p(u_\sharp^\star \mu, u_\sharp^\star \nu) \mathrm{d}\boldsymbol{\sigma}(u) \ , \tag{S27}$$

where $\boldsymbol{\sigma}$ is the uniform distribution on $\mathbb{S}^{d-1}$ and for any measurable function $f : \mathsf{Y} \to \mathbb{R}$ and $\zeta \in \mathcal{P}(\mathsf{Y})$, $f_\sharp \zeta$ is the push-forward measure of $\zeta$ by $f$. We approximate the integral in (S27) by selecting a finite set of projections $\mathsf{U} \subset \mathbb{S}^{d-1}$ and computing the empirical average:

$$\mathbf{SW}_p^p(\mu, \nu) \approx \frac{1}{\mathrm{card}(\mathsf{U})} \sum_{u \in \mathsf{U}} \mathbf{W}_p^p(u_\sharp^\star \mu, u_\sharp^\star \nu) \tag{S28}$$

The quality of this approximation depends on the sampling of $\mathbb{S}^{d-1}$. In our work, we use random samples picked uniformly on $\mathbb{S}^{d-1}$, as proposed in [12] and explained hereafter (see paragraph "Sampling schemes").

The Wasserstein distance between two one-dimensional probability densities $\mu$ and $\nu$ as defined in (6) is also estimated by replacing the integrals with a Monte Carlo estimate, and we can use two distinct methods to approximate this quantity.

The first approximation we consider is given by,

$$\mathbf{W}_p^p(\mu, \nu) \approx \frac{1}{K} \sum_{k=1}^{K} \left| \tilde{F}_\mu^{-1}(t_k) - \tilde{F}_\nu^{-1}(t_k) \right|^p \ , \tag{S29}$$

where $\{t_k\}_{k=1}^{K}$ are uniform and independent samples from $[0, 1]$ and for $\xi \in \{\mu, \nu\}$, $\tilde{F}_\xi^{-1}$ is a linear interpolation of $\bar{F}_\xi^{-1}$ which denotes either the exact quantile function of $\xi$ if $\xi$ is discrete, or an approximation by a Monte Carlo procedure. This last option is justified by the Glivenko-Cantelli Theorem.

The second approximation is given by,

$$\mathbf{W}_p^p(\mu, \nu) \approx \frac{1}{K} \sum_{k=1}^{K} \left| s_k - \tilde{F}_\nu^{-1}(\tilde{F}_\mu(s_k)) \right|^p \ , \tag{S30}$$

where $\{s_k\}_{i=1}^{K}$ are uniform and independent samples from $\mu$ and for $\xi \in \{\mu, \nu\}$, $\tilde{F}_\xi$ (resp. $\tilde{F}_\xi^{-1}$) is a linear interpolation of $\bar{F}_\xi$ (resp. $\bar{F}_\xi^{-1}$) which denotes either the exact cumulative distribution function (resp. quantile function) of $\xi$ if $\xi$ is discrete or an approximation by a Monte Carlo procedure.

**Sampling schemes:** We explain the methods that we used to generate i.i.d. samples from the uniform distribution on the $d$-dimensional sphere $\mathbb{S}^{d-1}$ and from multivariate elliptically contoured stable distributions.

- **Uniform sampling on the sphere.** To sample from $\mathbb{S}^{d-1}$, we form the $d$-dimensional vector $\mathbf{s}$ by drawing each of its $d$ components from the standard normal distribution $\mathcal{N}(0, 1)$ and we normalize it: $\mathbf{s}' = \mathbf{s}/\|\mathbf{s}\|_2$, so that $\mathbf{s}'$ lies on the sphere.

- **Sampling from multivariate elliptically contoured stable distributions.** We recall that if $Y \in \mathbb{R}^d$ is $\alpha$-stable and elliptically contoured, *i.e.* $Y \sim \mathcal{E}\alpha\mathcal{S}_c(\boldsymbol{\Sigma}, \mathbf{m})$, then its joint characteristic function is defined as, for any $\mathbf{t} \in \mathbb{R}^d$,

$$\mathbb{E}[\exp(i\mathbf{t}^T Y)] = \exp\left(-(\mathbf{t}^T \boldsymbol{\Sigma} \mathbf{t})^{\alpha/2} + i\mathbf{t}^T \mathbf{m}\right) \ , \tag{S31}$$

  where $\boldsymbol{\Sigma}$ is a positive definite matrix (akin to a correlation matrix), $\mathbf{m} \in \mathbb{R}^d$ is a location vector (equal to the mean if it exists) and $\alpha \in (0, 2)$ controls the thickness of the tail. Elliptically contoured stable distributions are scale mixtures of multivariate Gaussian distributions

[13, Proposition 2.5.2], whose densities are intractable, but can easily be simulated [14]: let $A \sim \mathcal{S}_{\alpha/2}(\beta, \gamma, \delta)$ be a one-dimensional positive $(\alpha/2)$-stable random variable with $\beta = 1$, $\gamma = 2\cos(\frac{\pi\alpha}{4})^{2/\alpha}$ and $\delta = 0$, and $G \sim \mathcal{N}(\mathbf{0}, \boldsymbol{\Sigma})$. Then, $Y = \sqrt{A}G + \mathbf{m}$ has (S31) as characteristic function.

**Optimization methods:** Computing the MSWE and MESWE implies minimizing the (expected) Sliced-Wasserstein distance over the set of parameters. In our experiments, we used different optimization methods as we detail below.

- **Multivariate Gaussian distributions.** We derive the explicit gradient expressions of the approximate $\mathbf{SW}_2^2$ distance with respect to the mean and scale parameters $\mathbf{m}$ and $\sigma^2$, and we use the ADAM stochastic optimization method with the default parameter settings suggested in [15]. For the MSWE, we use (S30) to approximate the one-dimensional Wasserstein distance, and we evaluate directly the Gaussian density of the generated samples, utilizing the fact that the projection of a Gaussian of parameters $(\mathbf{m}, \sigma^2\mathbf{I})$ along $u \in \mathbb{S}^{d-1}$ is a 1D normal distribution of parameters $(\langle u, \mathbf{m} \rangle, \sigma^2 \langle u, u \rangle)$. In this case, the gradient of the approximate $\mathbf{SW}_2^2$ between $\mu = \mathcal{N}(\mathbf{m}, \sigma^2\mathbf{I})$ and the empirical distribution associated to $n$ samples drawn by $\mathcal{N}(\mathbf{m}_\star, \sigma_\star^2\mathbf{I})$, denoted by $\hat{\nu}$, is given by,

$$\nabla_{\mathbf{m}}\mathbf{SW}_2^2(\mu, \hat{\nu}) = \frac{1}{\mathrm{card}(\mathsf{U})\,\mathrm{card}(\mathsf{S})} \sum_{u \in \mathsf{U}, s \in \mathsf{S}} \left( \left| s - \tilde{F}_{u_\sharp^\star \hat{\nu}}^{-1}(\tilde{F}_{u_\sharp^\star \mu}(s)) \right|^2 \mathcal{N}(s; \langle u, \mathbf{m} \rangle, \sigma^2 \|u\|^2) \right.$$

$$\left. \frac{s - \langle u, \mathbf{m} \rangle}{\sigma^2 \|u\|^2} u \right),$$

$$\nabla_{\sigma^2}\mathbf{SW}_2^2(\mu, \hat{\nu}) = \frac{1}{\mathrm{card}(\mathsf{U})\,\mathrm{card}(\mathsf{S})} \sum_{u \in \mathsf{U}, s \in \mathsf{S}} \left( \left| s - \tilde{F}_{u_\sharp^\star \hat{\nu}}^{-1}(\tilde{F}_{u_\sharp^\star \mu}(s)) \right|^2 \mathcal{N}(s; \langle u, \mathbf{m} \rangle, \sigma^2 \|u\|^2) \right.$$

$$\left. \frac{1}{2\sigma^2} \left( \frac{(s - \langle u, \mathbf{m} \rangle)^2}{\sigma^2 \|u\|^2} - 1 \right) \right),$$

where $\mathsf{U} \subset \mathbb{S}^{d-1}$ is a finite set of random projections picked uniformly on $\mathbb{S}^{d-1}$, $\mathsf{S}$ is a finite subset in $\mathbb{R}$, and for any $s \in \mathsf{S}$, $\mathcal{N}(s; \langle u, \mathbf{m} \rangle, \sigma^2 \|u\|^2)$ denotes the density function of the Gaussian of parameters $(\langle u, \mathbf{m} \rangle, \sigma^2 \|u\|^2)$ evaluated at $s$.

For the MESWE, we use (S29) and evaluate the empirical distribution of generated samples instead of their normal density. Therefore, the gradient of the approximate $\mathbf{SW}_2^2$ between the empirical distributions corresponding to one generated dataset of $m$ samples drawn from $\mathcal{N}(\mu, \sigma^2\mathbf{I})$ and $n$ samples drawn from $\mathcal{N}(\mu_\star, \sigma_\star^2\mathbf{I})$, respectively denoted by $\hat{\mu}$ and $\hat{\nu}$, is obtained with,

$$\nabla_{\mathbf{m}}\mathbf{SW}_2^2(\hat{\mu}, \hat{\nu}) = \frac{-2}{\mathrm{card}(\mathsf{U}).K} \sum_{u \in \mathsf{U}} \sum_{k=1}^{K} \left| \tilde{F}_{u_\sharp^\star \hat{\mu}}^{-1}(t_k) - \tilde{F}_{u_\sharp^\star \hat{\nu}}^{-1}(t_k) \right| u \ ,$$

$$\nabla_{\sigma^2}\mathbf{SW}_2^2(\hat{\mu}, \hat{\nu}) = \frac{1}{\mathrm{card}(\mathsf{U}).K} \sum_{u \in \mathsf{U}} \sum_{k=1}^{K} \left| \tilde{F}_{u_\sharp^\star \hat{\mu}}^{-1}(t_k) - \tilde{F}_{u_\sharp^\star \hat{\nu}}^{-1}(t_k) \right| \frac{\langle u, \mathbf{m} \rangle - \tilde{F}_{u_\sharp^\star \hat{\mu}}^{-1}(t_k)}{\sigma^2} \ .$$

- **Multivariate elliptically contoured stable distributions.** When comparing MESWE to MEWE, we approximate these estimators using the derivative-free optimization method Nelder-Mead (implemented in `Scipy`), following the approach in [6].

  When illustrating the theoretical properties of MESWE, we proceed in the same way as for the multivariate Gaussian experiment: we compute the explicit gradient expression of the approximate $\mathbf{SW}_2^2$ distance with respect to the location parameter $\mathbf{m}$, and we use the ADAM stochastic optimization method with the default settings. Equation (S32) gives the formula of the gradient of the approximate $\mathbf{SW}_2^2$ between the empirical distributions of one generated dataset of $m$ samples drawn from $\mathcal{E}\alpha\mathcal{S}_c(\mathbf{I}, \mathbf{m})$ and $n$ samples drawn from $\mathcal{E}\alpha\mathcal{S}_c(\mathbf{I}, \mathbf{m}_\star)$, respectively denoted by $\hat{\mu}$ and $\hat{\nu}$, with respect to $\mathbf{m}$.

$$\nabla_{\mathbf{m}}\mathbf{SW}_2^2(\hat{\mu}, \hat{\nu}) = \frac{-2}{\mathrm{card}(\mathsf{U}).K} \sum_{u \in \mathsf{U}} \sum_{k=1}^{K} \left| \tilde{F}_{u_\sharp^\star \hat{\mu}}^{-1}(t_k) - \tilde{F}_{u_\sharp^\star \hat{\nu}}^{-1}(t_k) \right| u \ . \tag{S32}$$

- **High-dimensional real data using GANs.** We use the ADAM optimizer provided by TensorFlow GPU.

**Computing infrastructure:** The experiment comparing the computational time of MESWE and MEWE was conducted on a daily-use laptop (CPU intel core i7, 1.90GHz × 8 and 16GB of RAM). The neural network experiment was run on a cluster with 4 relatively modern GPUs.

## Footnotes

[1] which exists since $f$ is assumed to have a compact support