[Reviews · NeurIPS 2019]

Reviewer 1



Clarity: the article is clear and well written, In this aspect the paper is an "accept" for me. (6) Significance: results are significant as they are new and they will generate some impact for practitioners. This is an accept as well (6) Quality: this paper is of high quality, it is clear there is a significant research effort behind. The combination "theoretical results + empirical validation in simple cases" is sensible given the type of paper this is, and the audience. Accept too (6) Originality: This is the item where I tend to reject more than to accept (5). I think it is definitely original, but all the theoretical contributions seem to me a bit marginal: I am very familiar with Bernton et al 2018, the paper that develops the technique (in turn, mainly based on Basseti et al 2006 and Pollard 1980) that is used here. After reading the supplement of this paper I am left with a "deja vu" feeling; some proofs look way too similar to Bernton et al, making me wonder whether they are rather straightforward adaptations. Indeed, It is no surprise all results of Bernton et al hold here too as this is a kind of average of one-dimensional wassersten distances. It is no surprise as well it is possible to establish a distributional limit in the multidimensional case here (unlike Bernton et al), although I think it is a very nice observation. Question for the authors: Does the obtained distributional limit here gives any intuition about what should be the limit in the multidimensional case for the vanilla wasserstein distance?

Reviewer 2



The paper is of high quality. It also comes with reasonable clarity. Overall it brings important new analysis to the important topic and the work seems to be original from what I can say.

Reviewer 3



***** After Author Response and Reviewer Discussions ***** I have gone through all the other reviews, the meta-reviewer's comment, and the authors' feedback. I will keep my evaluation unchanged. ********************************************************** *originality: To my best knowledge, the results are original. The methodologies of analysis belong to classical asymptotic statistics, but the problem analyzed is new. *quality: Due to the time constraint I did not go through the proofs. The claim in the abstract is well supported by the theorems. The work appears to be complete. The authors are honest about claiming the strength and weakness of their work. *clarity: The submission is clearly written, well written and of elegant style. Since this is a theoretical paper, the proofs provide enough information for an expert reader to varify the theorems which are the results. *significance: The results are important and significant for understanding the behavior of MSWE and MESWE. This paper is likely to be cited by people working on the application side of these two estimators. The problem is difficult and the authors have provided better analysis than that in literature to my best knowledge. The analysis provided advances the state of the art in a demonstrable way. The analysis is theoretically unique.

[Author Response · NeurIPS 2019]

We thank the reviewers for their useful feedback and their time. We are happy to see that all the reviews are positive and we appreciate that the reviewers found our article clear and of high quality.

We have corrected all the minor comments, as suggested. We now provide specific answers to each reviewer below.

**Reviewer #1:**

We thank the reviewer for their positive evaluation of our work and their comments.

*"All the theoretical contributions seem to me a bit marginal"* Since the Sliced-Wasserstein distance is an average of one-dimensional Wasserstein distances, the proofs for the existence, measurability and consistency of Sliced-Wasserstein estimators and the Central Limit Theorems (CLTs) are, indeed, inevitably similar to the proofs done in Bernton et al. [1]. However, the adaptation of these techniques was made possible by the derivation of novel properties regarding the topology induced by the Sliced-Wasserstein distance, which have never been investigated before and whose proofs differ from the ones in [1]: in Theorem 1, we show that the convergence in Sliced-Wasserstein implies weak convergence using non-standard techniques, and the same observation holds true for Lemma S6 (in the supplementary document), which establishes the lower semi-continuity of Sliced-Wasserstein. We believe that these two results are important contributions on their own and explain why the adaptation of [1] is not straightforward in the first place. Specifically, without Theorem 1, the formulation and use of Assumption A2 in our proofs would not make sense: see lines 174 and 221 in the supplementary document. We will explain these observations more explicitly to clarify our contributions.

On the other hand, we appreciate that the reviewer describes our Central Limit Theorems as *"a very nice observation"*. We think that it is another significant result compared to [1] since it applies to the multidimensional case, as the reviewer noticed. In their conclusion, [1] conjectures that the rate of the minimum Wasserstein estimators would "depend negatively on the dimension of the observation space rather than that of the parameter space", which suggests that the rate would suffer from the curse of dimensionality. Our result shows that it is not the case for the Sliced-Wasserstein distance. This curse of dimensionality has created a pessimism in the machine learning community about the use of Wasserstein-based methods in large dimensional settings (e.g. this fact has been popularly used for motivating regularized optimal transport). We believe that our findings provide an important counter-example to this conception and that the derived CLTs are thus another key contribution to the field. We will underline this observation in the paper.

*"Question for the authors: Does the obtained distributional limit here gives any intuition about what should be the limit in the multidimensional case for the vanilla wasserstein distance?"* This is a very interesting question. We believe it is definitely worth investigating whether our findings would help in the derivation of a CLT for the Wasserstein distance in the multidimensional case; however, we leave it out of the scope of this study.

**Reviewer #2:**

We thank the reviewer for the positive evaluation. We appreciate that they pointed out the importance of the problem and of our analysis given the rise in popularity of computational optimal transport.

*"The analysis is quite compactly presented in 10 pages and it could be useful to consider a longer version where the key findings are better elaborated in more details."* As suggested, we will add a new section to the supplementary document to further elaborate our theoretical findings.

**Reviewer #3:**

We thank the reviewer for giving a positive evaluation and a useful feedback. In particular, the reviewer noticed two minor problems:

*"Line 104, the second $\mathcal{P}_p$ should be $\mathcal{P}$"* We agree. We fixed this typo in the manuscript.

*"In thoerems 5 and 6, since the limit distribution is degenerated, why not claim the convergence as "in probability"?"* The limit distribution is in fact not degenerated since $G_\star$ is a random element: see Assumption A8. Therefore, we can not claim that the convergence in distribution derived in these theorems implies the convergence in probability. We will add further clarification in the manuscript to reflect this comment and avoid any confusion.

# References

[1] E. Bernton, P. E. Jacob, M. Gerber, and C. P. Robert. On parameter estimation with the Wasserstein distance. *Information and Inference: A Journal of the IMA*, Jan 2019.


[Meta-Review · NeurIPS 2019]

The reviewers liked the paper and voted for an accept that was confirmed following authors feedback. But the discussion highlighted the fact that the result do not discuss the problem of sampling on the unit sphere that needs to be done when actually learning generative models. It will probably add some variance in practice and should be at least discussed in the final paper and investigated in future works.